# Second-order forward-mode optimization of recurrent neural networks for neuroscience

**Youjing Yu**   **Rui Xia**   **Qingxi Ma**   **Máté Lengyel**   **Guillaume Hennequin**

Computational and Biological Learning Lab
Department of Engineering
University of Cambridge, Cambridge, UK
{yy471, rx220, qm218}@cam.ac.uk, {m.lengyel, g.hennequin}@eng.cam.ac.uk

## Abstract

Training recurrent neural networks (RNNs) to perform neuroscience tasks can be challenging. Unlike in machine learning where any architectural modification of an RNN (e.g. GRU or LSTM) is acceptable if it facilitates training, the RNN models trained as *models of brain dynamics* are subject to plausibility constraints that fundamentally exclude the usual machine learning hacks. The "vanilla" RNNs commonly used in computational neuroscience find themselves plagued by ill-conditioned loss surfaces that complicate training and significantly hinder our capacity to investigate the brain dynamics underlying complex tasks. Moreover, some tasks may require very long time horizons which backpropagation cannot handle given typical GPU memory limits. Here, we develop SOFO, a second-order optimizer that efficiently navigates loss surfaces whilst *not* requiring backpropagation. By relying instead on easily parallelized batched forward-mode differentiation, SOFO enjoys constant memory cost in time. Moreover, unlike most second-order optimizers which involve inherently sequential operations, SOFO's effective use of GPU parallelism yields a per-iteration wallclock time essentially on par with first-order gradient-based optimizers. We show vastly superior performance compared to Adam on a number of RNN tasks, including a difficult double-reaching motor task and the learning of an adaptive Kalman filter algorithm trained over a long horizon.

## 1   Introduction

In recent years, trained recurrent neural networks (RNN) have gained increasing adoption as models of brain circuits dynamics [47, 1]. As flexible parametric models of sequential dynamics, RNNs can be trained to perform specific computations or reproduce certain behaviors [31, 50, 20, 13], and be subsequently probed for insights into the distributed computations that give rise to those behaviors [2, 53]. RNNs are also often used as expressive models of the latent dynamics underlying the spatiotemporal structure of neural recordings [39, 44, 21, 12].

Despite their flexibility, RNNs are notoriously difficult to train [40, 51, 9, 36]. In addition to the classic problem of vanishing gradients in temporally extended tasks, the loss surfaces that arise from RNN dynamics in complex tasks often exhibit pathological curvature that first-order gradient-based optimization techniques struggle to handle. These problems are normally addressed in multiple ways. First, the RNNs used in machine learning are modified from their standard ("vanilla") formulation to include specific gate variables [18, 8] that largely mitigate the vanishing gradient problem. However, there is little biological support for the existence of such gating mechanisms, such that computational neuroscience studies have typically restricted themselves to vanilla RNNs [46]. Second, second-order optimizers such as Hessian-free optimization [32, 33, 50] or KFAC [34] can be used to dramatically accelerate training. However, those are difficult to scale, often much slower in wall-clock time, and

38th Conference on Neural Information Processing Systems (NeurIPS 2024).

memory hungry. Alternatively, the loss surface can be regularized by introducing artificial temporal "skip connections", and the network is progressively weaned off those biologically meaningless connections during the course of training [46].

Another great obstacle to training RNNs lies in the memory complexity of gradient-based optimization via backpropagation through time (BPTT). In certain applications, such as in motor neuroscience, RNNs must be trained on tasks that require the production of smooth behavior requiring the use of small (millisecond) time steps and therefore long time horizons. In other applications such as "learning to learn" [29, 10] where the parameters of a synaptic plasticity rule are optimized to yield a specific learning behavior, the network must be simulated over a horizon long enough to span the slow timescale of synaptic modifications. Since backpropagation has a memory cost that scales with the time horizon, its naive application is often prohibitively expensive in these scenarios. Memory efficiency can be improved in a variety of ways. Checkpointing [42] allows discarding information between sporadic checkpoints in the forward pass, at the expense of having to recompute that information during the backward pass. Alternatively, backpropagation through time can be truncated [51]: instead of computing a single gradient for the entire sequence of size $T$, gradients are computed sequentially for consecutive sub-sequences of size $T' \ll T$. Although cutting the chain of backpropagating gradients in this way reduces the memory cost from $\mathcal{O}(T)$ to $\mathcal{O}(T')$, the resulting gradient estimates are biased and neglect long-range temporal dependencies that may be important for the task [35, 38]. Yet another way of reducing the memory footprint of BPTT is to formulate the RNN dynamics as an ordinary differential equation (ODE) in continuous time. In these "neural ODE" models, gradients are obtained by solving another (adjoint) ODE with no need for caching intermediate states [7, 26]. However, this requires very accurate, often adaptive ODE solvers that are highly non-trivial to implement and to adapt to the mini-batch setting.

Finally, training RNNs is known to largely under-exploit the parallelization capabilities of modern (GPU) hardware. RNN gradient computations are inherently sequential (forward-backward), such that the only way to exploit GPU parallelism is to use large batches (e.g. run a large number of trials of the task in each training iteration). Whilst using large batches may enable faster learning with larger learning rates, it is often the case that increasing the batch size beyond a certain point no longer helps in this respect [25]. Thus, some of the parallel processing power of GPU hardware is often underused.

In summary, training RNNs is plagued with many problems. While there exist piecemeal solutions to each of these challenges, we currently lack a simple method that addresses all of them simultaneously. In this paper, we develop such a method, which we call SOFO (Second order Forward-mode Optimization). At each training iteration, SOFO reparameterizes the model in a random subspace, and uses batched forward-mode automatic differentiation to efficiently compute exact Generalized Gauss-Newton (GGN) updates in that subspace. This circumvents the need for backpropagation, enabling the training of RNNs over very long horizons. We test the performance of SOFO on benchmark tasks (e.g. learning an adaptive Kalman filter over long horizons, performing a motor reach with a biomechanical arm; Section 4). We show empirically that, despite only exploring a small fraction of the parameter space in each iteration, SOFO dramatically accelerates training of RNNs in these complex tasks. More generally, we suggest that SOFO – as a general purpose optimizer – is a strong candidate for machine learning applications that involve relatively few trainable parameters but memory-hungry computational graphs(e.g. fine-tuning of transformers using low-rank adapters [19], or tuning of hyperparameters in Gaussian process-based models).

## 2 Background and related work

### 2.1 Problem formulation

We consider supervised learning problems where a neural network with parameters $\boldsymbol{\theta} \in \mathbb{R}^P$ is given batches of inputs $\boldsymbol{x}$, and produces batches of outputs $\boldsymbol{y}(\boldsymbol{x}, \boldsymbol{\theta}) \in \mathbb{R}^{MN}$, where $M$ is the batch size. The specific case of an RNN with output dimension $D$ simulated over a time horizon $T$ would imply $N = DT$ – i.e. the network's outputs consist of entire sequences. Training the network involves minimizing a stochastic cost function $c(\boldsymbol{\theta}, \boldsymbol{x})$, averaged over the training data in each minibatch, that

depends on the network's output $\boldsymbol{y}$ and therefore on the parameters $\boldsymbol{\theta}$:

$$c(\boldsymbol{\theta}, \boldsymbol{x}) \triangleq \ell(\boldsymbol{y}(\boldsymbol{\theta}, \boldsymbol{x})) \triangleq \frac{1}{M} \sum_{m=1}^{M} \sum_{i=1}^{N} \ell_{mi}\left(\boldsymbol{y}_{mi}(\boldsymbol{\theta}, \boldsymbol{x}_m)\right) \tag{1}$$

where each $\ell_{mi}(\cdot)$ function is strongly convex in its first argument (which does not imply that the overall cost $c(\boldsymbol{\theta}, \cdot)$ is convex in $\boldsymbol{\theta}$). In RNN applications, $i$ runs from 1 to $N = DT$, i.e. it indexes the cartesian product of $T$ time bins $\times$ $D$ output dimensions; the explicit dependence of $\ell_{mi}$ on $i$ thus allows for the possibility of providing teaching signals only at specific times. To eliminate clutter, in most of this paper we will drop the explicit dependence of the network's outputs on the inputs $\boldsymbol{x}$, focusing instead on their dependence on $\boldsymbol{\theta}$.

## 2.2 Generalized Gauss-Newton optimization

The Generalized Gauss-Newton (GGN) method prescribes parameter updates of the form:

$$\boldsymbol{\theta}_{t+1} = \boldsymbol{\theta}_t - \eta\, G(\boldsymbol{\theta}, \boldsymbol{x})^+ \left.\frac{\partial c}{\partial \boldsymbol{\theta}}\right|_{\boldsymbol{\theta}_t} \tag{2}$$

where $\eta$ is a learning rate and $G(\boldsymbol{\theta}, \boldsymbol{x})^+$ is the Moore-Penrose pseudo-inverse of the GGN matrix $G(\boldsymbol{\theta}, \boldsymbol{x}) \in \mathbb{R}^{P \times P}$, defined as

$$G(\boldsymbol{\theta}, \boldsymbol{x}) \triangleq J(\boldsymbol{\theta}, \boldsymbol{x})^\top \left(\left.\frac{\partial^2 \ell}{\partial \boldsymbol{y}^2}\right|_{\boldsymbol{y}(\boldsymbol{\theta}, \boldsymbol{x})}\right) J(\boldsymbol{\theta}, \boldsymbol{x}). \tag{3}$$

Here, $J(\boldsymbol{\theta}, \boldsymbol{x}) \in \mathbb{R}^{MN \times P}$ is the Jacobian of the network's output $\boldsymbol{y}$ w.r.t. the parameters $\boldsymbol{\theta}$. We note that for an underparameterized model with a number of network outputs $MN$ exceeding the number of parameters $P$, the GGN matrix can be full rank, in which case the pseudoinverse coincides with the inverse. For overparameterized models, it has rank $\leq MN$. In both cases, it is common practice to damp the inverse to enhance the stability of the training process; this is done by using $G_\gamma^+ \triangleq (G + \gamma I)^{-1}$ instead of $G^+$ with some damping parameter $\gamma$.

## 2.3 Forward-mode automatic differentiation

Given a computational graph with inputs $\boldsymbol{\theta} \in \mathbb{R}^P$ (keeping the notation relevant to our use case) and producing outputs $\boldsymbol{y}(\boldsymbol{\theta})$, forward-mode AD [3] allows the directional derivative $\frac{\partial \boldsymbol{y}(\boldsymbol{\theta})}{\partial \boldsymbol{\theta}} \boldsymbol{v}$ (i.e. a Jacobian-vector product, JVP) in any arbitrary direction $\boldsymbol{v}$ to be automatically computed together with $\boldsymbol{y}(\boldsymbol{\theta})$ itself, with roughly the same computational and memory complexity. This is achieved by initializing the parameters with primal value $\boldsymbol{\theta}$ and tangent value $\dot{\boldsymbol{\theta}} = \boldsymbol{v}$, extending the standard mathematical operators to operate on such primal/tangent pairs, and running the computation forward in this dual space. For any intermediate node $\boldsymbol{z}$ in the computational graph, the tangent value $\dot{\boldsymbol{z}}$ that is produced has the same shape as $\boldsymbol{z}$ and represents the sensitivity of $\boldsymbol{z}$ to small one-dimensional changes in $\boldsymbol{\theta}$ in the direction of $\boldsymbol{v}$:

$$\dot{\boldsymbol{z}} \triangleq \lim_{\epsilon \to 0} \frac{\boldsymbol{z}(\boldsymbol{\theta} + \epsilon \boldsymbol{v}) - \boldsymbol{z}(\boldsymbol{\theta})}{\epsilon} \quad = \quad \underbrace{\frac{\partial \boldsymbol{z}}{\partial \boldsymbol{\theta}}}_{\mathbb{R}^{\bullet \times P}} \underbrace{\boldsymbol{v}}_{\mathbb{R}^P}. \tag{4}$$

Note that – unlike backpropagation, or "reverse-mode AD" – forward-mode AD does not require caching any of the intermediate results that lead to the output of interest. SOFO exploits the fact that Jacobian-vector products are embarassingly parallelizable, i.e. one can rewrite the standard maths functions to operate not on one tangent per value, but on a whole batch of $K$ tangents in parallel. This can be done in JAX [5] out-of-the-box by composing Jacobian-vector products with the `vmap` primitive. As no such functionality exists in PyTorch yet, we provide our own flexible implementation of batched JVPs based on OCaml-Torch[1].

---

[1] code available at `https://github.com/hennequin-lab/SOFO`

## 2.4 Subspace optimization methods

A variety of methods exist that, like SOFO, optimize parameters in a different subspace in each iteration. A well-known example is coordinate descent, which iteratively minimizes the objective function w.r.t. each parameter, one at a time [55]. More generally, one may optimize in a higher-dimensional subspace than 1D, in a coordinate system that is not necessarily axis-aligned. This leads to a family of randomized subspace algorithms, including stochastic subspace descent (SSD; 27) and its variance-reduced version inspired by stochastic variance-reduced gradient (SVRG; 22). The above methods can be construed as Jacobian sketching methods, whereby a random "sketch" of the Jacobian is obtained and used to estimate the gradient; the convergence of such sketching algorithms has been proved [27, 28] under standard assumptions on the loss function (e.g. Polyak-Lojasiewicz condition; 24). Similarly, it is possible to sketch not only the Jacobian but also any of the matrices that are normally used as curvature estimates, leading to a family of randomized second-order optimization algorithms. These include the sketched Newton algorithm [41], randomized subspace Newton [14], randomized subspace Gauss-Newton [6], stochastic dual Newton ascent [43] and stochastic subspace cubic Newton [15]. Sketched Hessians are usually obtained by nested forward- and reverse-mode AD, and as such incur the same memory complexity as backpropagation. To obtain a rank-$K$ sketch of the Jacobian, and therefore compute directional first-order derivatives in a $K$-dimensional subspace, one can instead perform $K$ independent forward-mode AD computations [28, 4].

Here, we extend this use of forward-mode AD to the sketching of the Generalized Gauss-Newton matrix, which we show can be performed efficiently on GPUs. This leads to a memory- and compute-efficient algorithm that enjoys the fast convergence properties of a second-order method, with the runtime complexity of a first-order optimizer.

## 3  SOFO

SOFO (Algorithm 1) is based on successively optimizing low-dimensional affine re-parameterizations of the model, randomized in each training iteration. Specifically, instead of updating all $P$ parameters *independently* at every step as is normally done, SOFO locally reparameterizes the model by writing the cost function as

$$\tilde{c}_t(\Delta\tilde{\boldsymbol{\theta}}) \triangleq c(\boldsymbol{\theta}_t + \Theta\Delta\tilde{\boldsymbol{\theta}}), \tag{5}$$

where the $K$ columns of $\Theta$ form a random $K$-dimensional subspace that is drawn anew at every iteration. An exact GGN update $\Delta\tilde{\boldsymbol{\theta}}^\star$ (see below) is then obtained for this momentary lower-order model, leading to new parameters $\boldsymbol{\theta}_{t+1} = \boldsymbol{\theta}_t - \eta\,\Theta\Delta\tilde{\boldsymbol{\theta}}^\star$ around which the model is again re-parameterized in the next iteration. The gradient and GGN matrix associated with $\tilde{c}_t(\cdot)$ are related to

---

**Algorithm 1** SOFO: Second-order Forward Optimisation

1: **input:** $\boldsymbol{\theta}_0 \in \mathbb{R}^P$
2: **hyperparameters:** subspace dimension $K$, learning rate $\eta$, relative damping $\lambda$
3: **convention:** for any variable $\boldsymbol{z} \in \mathbb{R}^\bullet$, let uppercase $Z \in \mathbb{R}^{\bullet \times K}$ denote the associated batch of $K$ tangent values (e.g. $\{\boldsymbol{\theta}, \Theta\}$, $\{\boldsymbol{y}, Y\}$, $\{c, C\}$)
4: **for** $t = 0, 1, 2, \dots$ **do**                                                                   ▷ training iterations
5:     sample data minibatch $\boldsymbol{x}$
6:     sample $\Theta \in \mathbb{R}^{P \times K}$ with $\Theta_{pk} \overset{\text{iid.}}{\sim} \mathcal{N}(0, 1)$                          ▷ subspace randomization
7:     $\{\boldsymbol{y}, Y\} = \texttt{network\_output}(\{\boldsymbol{\theta}, \Theta\}, \boldsymbol{x})$           ▷ under batched forward-mode AD
8:     $\{c, C\} = c(\{\boldsymbol{y}, Y\})$ (c.f. Equation 1)              ▷ under batched forward-mode AD
9:     form sketched GGN matrix $\tilde{G} = Y^\top H Y$                                    ▷ c.f. Equation 7
10:    compute SVD of $\tilde{G} = USU^\top$
11:    extract $s_{\max} = \max \text{diag}(S)$                                          ▷ max. singular value
12:    $\boldsymbol{\theta}_{t+1} = \boldsymbol{\theta}_t - \eta\,\Theta U(S + \lambda s_{\max}I)^{-1}U^\top C$                ▷ SOFO update
13: **end for**
14: **output:** last iterate $\boldsymbol{\theta}_t$
15: **note:** for RNNs, memory can be saved in steps 7–9 by directly accumulating $\tilde{G}$ and $C$ over time steps.

---

those of $c(\cdot)$ via the chain rule:

$$\underbrace{\frac{\partial \tilde{c}_t}{\partial \Delta \tilde{\boldsymbol{\theta}}}}_{\mathbb{R}^K} = \Theta^\top \left. \frac{\partial c}{\partial \boldsymbol{\theta}} \right|_{\boldsymbol{\theta}_t} \quad \text{and} \quad \underbrace{\tilde{G}}_{\mathbb{R}^{K \times K}} = \Theta^\top G \Theta. \tag{6}$$

Thus, $\tilde{G}$ is a random $(K \times K)$ sketch of the full $(P \times P)$ GGN matrix (c.f. Section 2.4). Inserting Equation 3 into Equation 6, we obtain

$$\tilde{G} = (J\Theta)^\top \left( \frac{\partial^2 \ell}{\partial \boldsymbol{y}^2} \right) (J\Theta) \tag{7}$$

which is then used to compute the main (damped) SOFO parameter update,

$$\boldsymbol{\theta}_{t+1} = \boldsymbol{\theta}_t - \eta \, \Theta (\tilde{G} + \gamma I)^{-1} \left( \Theta^\top \frac{\partial c}{\partial \boldsymbol{\theta}} \right), \tag{8}$$

where $\gamma$ is the damping parameter (see below). SOFO uses batched forward-mode AD to efficiently compute $J\Theta$ and $\Theta^\top \frac{\partial c}{\partial \boldsymbol{\theta}}$ on parallel hardware. As in standard forward AD, we rewrite the standard math operators to operate on primal/tangent pairs (c.f. Section 2.3). However, instead of working with single tangents, we operate directly on entire batches of $K$ tangent values for every primal value (note: these tangent batches have nothing to do with data minibatches). Those $K$ tangents can be propagated completely independently of each other through every differentiable operation leading to network outputs and the final cost. Thus, computing such Jacobian-matrix products is an embarrassingly parallel problem that can fully exploit GPU parallelism. The steps detailed in Algorithm 1 show concretely how $J\Theta \equiv Y$ is obtained as the tangent batch associated with the network outputs $\boldsymbol{y}$, and how the vector of directional derivatives $\Theta^\top \frac{\partial c}{\partial \boldsymbol{\theta}} \equiv C$ is the tangent batch associated with the final cost $c$.

Given that $\boldsymbol{y}$ is typically a large tensor (batch size $\times$ time horizon $\times$ output dimension), one might worry that the Hessian of $\ell(\boldsymbol{y})$ in . Equation 7 could be expensive to compute. However, (i) the loss function $\ell(\boldsymbol{y})$ is typically a sum over losses applied to individual batch elements and time bins of $\boldsymbol{y}$, implying a block-diagonal Hessian and therefore cheap products with $J\Theta$; (ii) for standard losses such as the mean squared error or the cross-entropy, the diagonal blocks themselves have a structure that affords fast computations [32]. One might also worry that $\boldsymbol{y}$ may not even fit in GPU memory. For RNNs, however, $\ell(\cdot)$ is typically a sum of losses accumulated over time, and it is straightforward to similarly accumulate both the overall cost $c$ (and its tangents) and the sketched GGN matrix $\tilde{G}$ without ever having to store network activations over time. Overall, this leads to a $\mathcal{O}(T^0)$ memory cost (Figure 6A).

Second-order methods are known to require appropriate damping of the curvature matrix [33]. Finding a good absolute damping parameter $\gamma$ can be difficult without knowing the overall scale of the curvature matrix or the extent of its ill-conditioning. SOFO's lower-dimensional reparameterization of the model at each iteration affords us the explicit computation and inversion of the sketched GGN matrix $\tilde{G}$ $(K \times K)$. In particular, this gives us easy access to its singular values. We exploit this here by using a *relative* damping scheme, setting $\gamma = \lambda s_{\max}$ where $s_{\max}$ is the maximum singular value of $\tilde{G}$, and $\lambda$ is a relative damping parameter that we have found easier to tune.

**Connection to real-time recurrent learning (RTRL)** The RTRL algorithm [54] is mathematically equivalent to a particular limit of SOFO: the limit where (i) SOFO doesn't use curvature information (i.e. dropping the inverse matrix term in Equation 8 ) and (ii) the set of tangent vectors used at each iteration ($\Theta$ in Equation 6) is a full basis equal to the identity matrix. In this limit, batched forward-mode differentiation (efficiently) implements the usual RTRL recursion to propagate exact, entire Jacobians of network activity w.r.t. the parameters, in (constant memory) forward mode. For a model with $P$ parameters, the memory cost of RTRL will be $P/K$ times that of SOFO if SOFO uses $K$ random tangents. Given that the ratio $K/P$ used in our experiments is around $1\%$, this would make RTRL $\sim 100$ times more memory-intensive than SOFO.

## 4 Results

We now apply SOFO to a range of RNN-based applications relevant to neuroscience, and show that it outperforms Adam in all cases, occasionally finding network solutions for tasks where Adam failed.

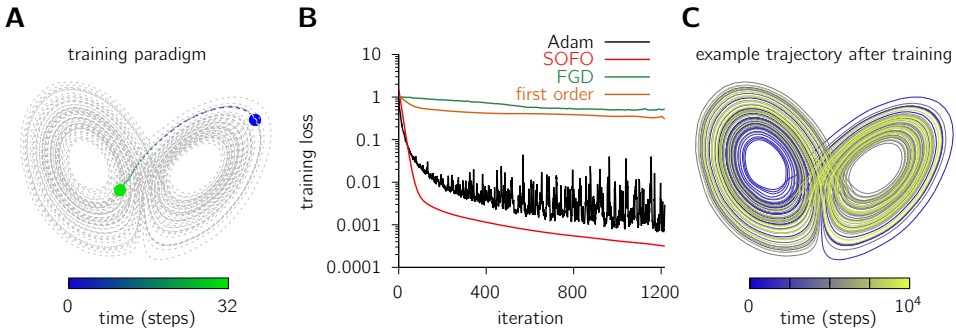

Figure 1: **Sparsely supervised Lorenz attractor**. (**A**) In each trial, a 32-step snippet (colored line) of the Lorenz manifold attractor (grey) is chosen at random. The RNN is initialized in the starting state (blue dot), and the final state (green dot) is the only supervision label provided during training. (**B**) Learning curves for Adam, SOFO, FGD [4] and a first-order version of SOFO that uses the same number of tangents [28]. It is possible to get a smoother learning curve for Adam but at the cost of much slower convergence. (**C**) Example trajectory produced by the RNN after training with SOFO. In this task, SOFO is only 35% slower than Adam in wallclock time per iteration (not shown).

For all tasks, we manually tuned both SOFO and Adam's hyperparameters to get best performance in each case, as assessed by the training loss. All experiments are run on an RTX 2080 Ti GPU.

### 4.1 Inferring dynamics from sporadic observations

We begin by evaluating SOFO's ability to learn the nonlinear dynamics of a system based on temporally sparse state observations – a problem related to current neuroscience efforts to infer latent brain dynamics from neural or behavioral data [12]. We use the Lorenz attractor [30] as a model nonlinear system with non-trivial dynamics (Figure 1). We train an RNN with a 3-dimensional state space, but flexible one-step dynamics parameterized by a two-layer MLP with an inverted bottleneck (Appendix E.1; this is similar to the neural ODE model class used in 26). In each trial, the network is initialized at a random location on the Lorenz manifold, and is trained to generate a state trajectory that, after 32 time steps, terminates exactly where the Lorenz system would have (Figure 1A). Therefore, the RNN is only supervised using trajectory endpoints, being left unsupervised for most of the trial.

In this task, the training loss decays faster, more smoothly, and to a lower minimum with SOFO than with Adam (Figure 1B). The RNN correctly recovers the Lorenz dynamics (Figure 1C).

It is remarkable that despite only optimizing a small effective fraction $K/P$ of the parameters at each iteration (in this application, $K = 128$ implies $K/P < 5\%$), SOFO still outperforms a method that optimizes all parameters at once (Adam). We hypothesized that this owes to the use of second-order preconditioning with the GGN matrix. To test this, we compared SOFO to a first-order version of it, i.e. parameter updates of the form $\Theta\Theta^\top \frac{\partial c}{\partial \theta}$ [28]. This first-order subspace method also sees the same small parameter subspace at each iteration, but does much worse than SOFO; in fact, in this case, it appears to be useless (Figure 1B, orange). With $K = 1$ and no second-order preconditioning, SOFO reduces to "Forward-gradient descent" (FGD; 4), which performs even worse (Figure 1B, green).

### 4.2 Learning an adaptive Kalman filter

Next, we trained a vanilla RNN to perform adaptive Kalman filtering (KF) in a non-stationary environment (Figure 2A) – a task highly relevant to adaptive motor control [16, 17]. In this task, the RNN receives noisy observations of the state of an underlying 1D linear dynamical system (LDS), and at each time step must infer the current latent state. The parameters of the linear dynamical system (i.e. the context) are subject to sporadic changes during each trial (details in Appendix E.2). Thus, the RNN must learn to integrate its inputs to (i) rapidly learn about the current parameters of the LDS to be able to perform optimal KF, and (ii) detect any contextual switches.

SOFO again outperforms Adam on this task, successfully training the RNN in less than 200 training iterations (Figure 2B). Adam is much slower, and converges to a suboptimal solution. Networks

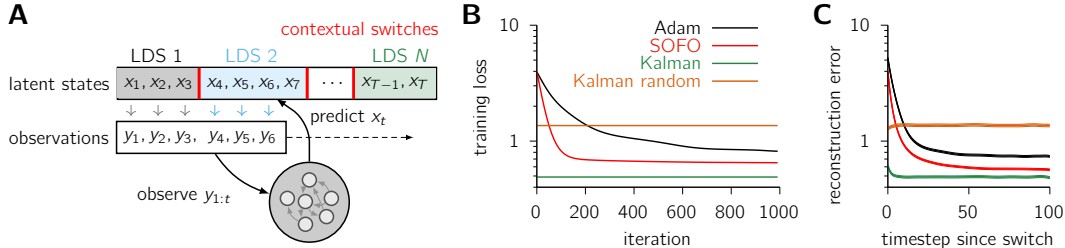

Figure 2: **Learning an adaptive Kalman filter (KF)**. (**A**) Task structure, where $x_t$ denotes the latent state and $y_t$ its momentary observation. Each trial ($T = 5000$ steps) is randomly partitioned into successive contexts (of potentially different durations; Appendix E.2). Each context is characterized by a different, randomly parameterized LDS, which produces latent state trajectories noisily observed by the RNN; context switches are uncued. At each step, the RNN must predict the current state. (**B**) Training curves for Adam and SOFO, compared to the MSE noise floor provided by KF knowing the current context (green, average over 1000 trials), and a baseline showing KF performance given random (and thus wrong) LDS parameters sampled from the task distribution (Kalman random; orange, average over 1000 trials). (**C**) Within-trial evolution of the mean-squared prediction error for the current context in the trained models, as a function of time elapsed since the last context switch, averaged over 1500 contexts (barely visible shaded areas $= \pm 2$ s.e.m.). Green and orange baselines: same as in (B).

trained by SOFO are able to (implicitly) infer the current LDS parameters with good accuracy within 10/20 steps following every context switch, gradually approaching the fundamental limit set by a KF that has full knowledge of the active LDS parameters at all times (Figure 2C, red). Within-context learning in networks trained by Adam is slower and worse overall (black).

### 4.3   3-bit flip-flop task

To further demonstrate the ability of SOFO to learn long temporal dependencies, we turn to the 3-bit flip-flop task described in [49] (Figure 3A, B; see Appendix E.3 for details). Moreover, we compare the performance of SOFO not only against Adam, but also FORCE [48] (Figure 3C), another second-order optimizer based on recursive least-squares (RLS) which is admittedly the optimal solution in the "reservoir" setting. Whilst FORCE does make more rapid initial progress, SOFO eventually achieves a lower loss at convergence. To make the comparison fair to FORCE, we implemented a novel batched version of FORCE-RLS (Appendix B).

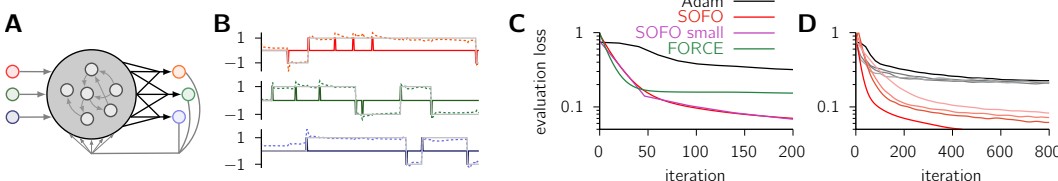

Figure 3: **3-bit flip-flop task**. (**A**) Network architecture for the classical 3-bit flip flop task: three input channels sporadically provide random bits ($\pm 1$) at random time intervals and each corresponding output channel must hold its activity at the value given by the last provided bit (see **B**). The network feeds back its own output via random feedback weights (grey). Only the output weights (black) are trained. (**B**) Example outputs for the network trained with SOFO (dashed lines); tight match to target outputs (solid pale) shows successful training. Solid dark lines show the corresponding input bits. (**C**) Evaluation loss for Adam (black), SOFO (red) and FORCE (green), for a network of size 1000 and for SOFO small (purple) where the network size is reduced to 128 but the all sets of weights (i.e. recurrent weights, biases etc) are trainable. (**D**) Same as C, for varying number of neurons from 1000 to 8000 (dark to pale).

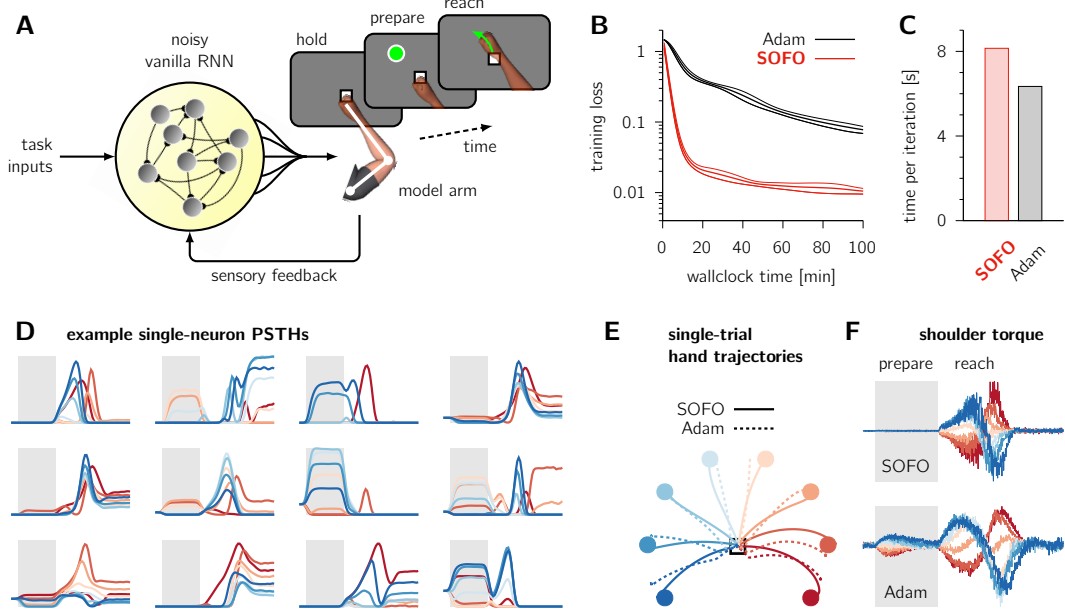

Figure 4: **Memory-guided single-reach task**. **(A)** System schematics and task structure (see text). **(B)** Training curves for Adam and SOFO (mean $\pm$ 2 s.e.m. over 3 independent runs). **(C)** Wallclock time per training iteration. **(D–F)** Example single-neuron firing rates (D), single-trial hand trajectories (E) and shoulder torques (F) for eight selected reach conditions (color-coded). Gray areas denote preparation epochs.

Interestingly, we find that SOFO works best when the reservoir is non-chaotic at initialization ($g = 0.5$ instead of $g = 1.5$ as in [49]) – whereas FORCE tends to fail in this setting (as is well known). Morover, ref. [11] demonstrated that by restricting the locus of learning to those few task-relevant output weights, FORCE tends to come up with brittle solutions. SOFO, on the other hand, is a flexible general purpose $2^{\text{nd}}$-order optimizer that can be used to tweak any of the network's parameters (recurrent weights, biases, feedback and input weights), not just the output weights. We implemented this form of training and found that SOFO was able to train a much smaller network of 128 units and ReLu activations (instead of the usual FORCE tanh) on the 3-bit flip flop task (Figure 3C, purple), to a final test means squared error more than 15 times smaller than FORCE's with 1000 neurons. In summary, by allowing efficient training of more flexible network models from non-chaotic initializations, SOFO stands as nicely complementary to FORCE in the RNN training toolbox.

## 4.4 Motor tasks

Finally, we also applied SOFO to two different memory-guided motor tasks: a single-reach task, and a more challenging double-reach task. In both tasks, the RNN receives task-related inputs (see below) and outputs a pair of time-varying torques driving the motion of a two-jointed model arm (Figure 4A; [23]). The RNN also receives sensory feedback in the form of joint positions and velocities. In the single-reach task (Figure 4), each trial begins with a variable "idle" phase during which the hand must remain at a central spot (where it is initialized). A variable-length preparation phase follows, during which the network is presented with inputs representing the $x$- and $y$-coordinates of the reach target but must continue to hold to the central spot. The movement phase is initiated by the withdrawal of a "hold" input present during the first two phases (c.f. [50]). The RNN must reach the target within 600 ms, and hold it for 200 ms. In the double-reach task, the two target locations are simultaneously presented during the preparation phase, and must be reached in a sequence (see Section 4.4 for details of the loss function). Target locations are randomly sampled in each trial from a distribution of reachable positions around the central spot.

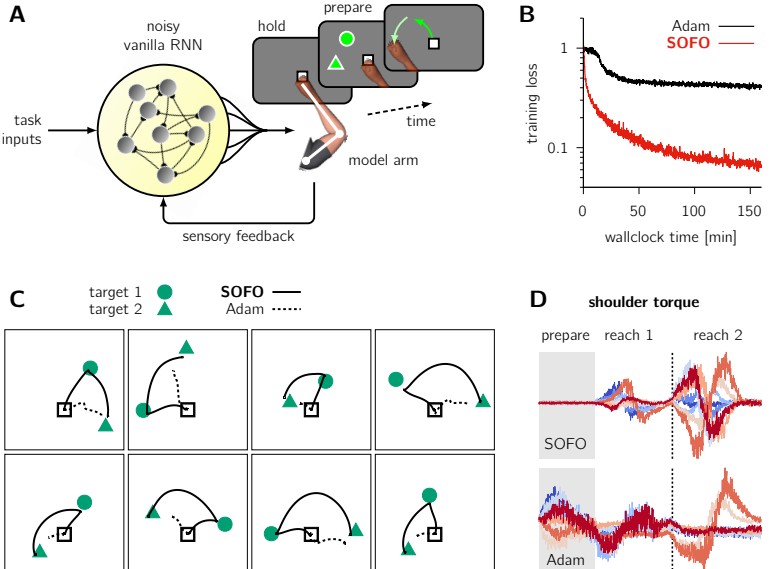

Figure 5: Memory-guided double-reaching task. **(A)** Task structure (see text). **(B)** Training curves for Adam and SOFO. **(C)** Example single-trial hand trajectories for selected pairs of consecutive reach targets. **(D)** Example time series of shoulder torques.

We trained a *stochastic* vanilla RNN on both tasks. This is a difficult task for a noisy RNN to acquire, as process noise in the recurrent dynamics not only causes motor disturbances in the arm, but also corrupts the memory of the target location which is no longer visible during the reach. A successful network must learn a strategy that mitigates the effect of ongoing noise in the relevant memory / output subspaces, and integrate sensory feedback to correct for output disturbances.

SOFO rapidly trains successful RNNs to perform both tasks proficiently in a few hundred training iterations (between 30min and 2h of wallclock time), where RNNs trained using Adam just about get the gist of the single-reach, and completely fail to learn double reaching (Figure 4B-E-F and Figure 5B-C-D) despite days of manual hyperparameter tuning. In the single-reach task, networks trained with Adam fail to withhold movement until the go cue, and appear unable to robustly memorize the target location and/or correct for motor noise.

Excitingly, the single-neuron activity patterns observed in the RNN trained with SOFO have a striking resemblance with those recorded in the motor cortex of non-human primates during similar tasks. Some neurons are active during movement only, producing rich multiphasic activity patterns; others are active during preparation only, and many are active during both phases. We leave the analysis of these population activity trajectories to future work; for now, we simply note that SOFO renews our ability to train networks on complex motor tasks, thus opening many avenues for future motor neuroscience investigations.

### 4.5 Memory and compute profiling

Details on SOFO's memory and algorithmic complexity can be found in Appendix C. To experimentally demonstrate that SOFO is memory and compute efficient, we carried out profiling in the context of the Kalman filter learning task (Section 4.2), varying the time horizon $T$ used in each trial. As expected from backpropagation, memory usage for Adam increases almost linearly with the time horizon, eventually exceeding the limit of our GPU for $T \sim 30K$ steps (Figure 6A, black). In contrast, SOFO's use of forward-mode AD incurs a low and constant memory cost independent of $T$ (red). Remarkably, despite being a second-order method, SOFO is only about twice slower than Adam on wall-clock time per iteration (Figure 6B; see also wallclock time comparisons in Figure 4B and Figure 5B).

## 5   Limitations of SOFO

All tasks we have used here can be described using a relatively small number of bits – much fewer than the information that could in principle be stored in the model's parameters. Accordingly, simply

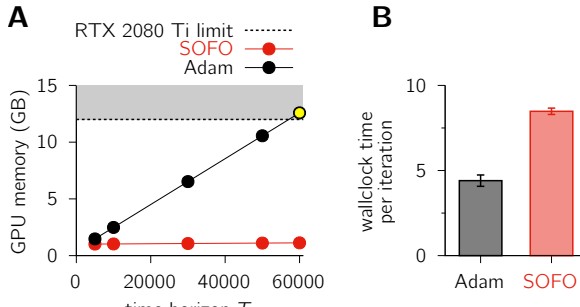

Figure 6: **Compute time and memory profiling.** (**A**) GPU memory usage as a function of time horizon used in the Kalman filter task (Section 4.2). Gray shading indicate the 12GB memory limit of an RTX 2080 Ti. (**B**) Wallclock time per iteration (mean $\pm$ s.e.m.), for $K = 256$, $T = 5000$, and a batch size of 256. GPU memory usage is taken from `nvtop`.

instating the right low-dimensional dynamical motifs in the RNN is sufficient to acquire the desired behaviour [9]. Therefore, it is perhaps not surprising that SOFO performs very well even though it only optimizes a fraction of all model parameters in each iteration. We speculate that in richer learning tasks that require the RNN to store a lot of information (e.g. supervised learning on a rich dataset), SOFO might not have as large an advantage over Adam. Indeed, our experiments using SOFO to train an MLP-Mixer [52] on CIFAR-10 shows relatively poor performance – at best on par with Adam (Appendix D.3).

Even for simple tasks, and assuming a fixed $K$ (size of the gradient and GGN sketches), we expect SOFO's performance to gradually deteriorate with the number of parameters in the model. This is because the sketch of the GGN matrix gets noisier as the tangent to parameter ratio ($K/P$) gets smaller – meaning that we explore a smaller fraction of the parameter space per iteration. Indeed, when training RNNs with increasing number of neurons $S$, the convergence rate for SOFO worsens while that for Adam improves (Appendix D.2).

## 6 Conclusion and Future Work

In this paper, we have shown how sketches of both the Generalized Gauss-Newton matrix and the loss gradient can be computed efficiently on parallel hardware, leading to an RNN optimizer that outperforms Adam in all the neuroscience-related tasks we have studied, at lower memory complexity. By accelerating RNN training (and sometimes enabling successful training altogether), SOFO could greatly facilitate a whole line of neuroscientific inquiry that relies on constructing neural networks that solve behavioral tasks. These trained networks are constrained for biological realism and reverse-engineered to increase our understanding of brain computations. SOFO's fast and robust convergence in all the tasks we have tried suggests that it could enable faster analysis-by-synthesis iterations.

Our current implementation of SOFO could be refined in a number of ways. Adaptive damping (e.g. based on the Levenberg-Marquardt heuristic [37, 33]) is known to make second-order optimization algorithms more robust. We speculate that SOFO might benefit from it – although perhaps not to the same usual degree, given that a random sketch of the GGN matrix is typically better-conditioned than the full GGN (it is indeed very unlikely that parameter subspaces randomly sampled at each iteration would exactly contain the top and bottom eigenvectors of the GGN).

In principle, SOFO is a general-purpose optimizer that could be deployed onto any problem outside the realm of recurrent neural networks. We expect SOFO to perform particularly well in models that have relatively few tunable parameters yet involve computational graphs that are too large to allow backpropagation to operate on large data batches. A prime example of this is the fine tuning of large language models, e.g. with low-rank adaptors [19], but there are many other potential applications e.g. in the physical sciences.

## Acknowledgments

This work was funded by Trinity College (graduate studentship to Y.Y.), EPSRC / MediaTek Research UK (iCase studentship to R.X.) and the Wellcome Trust (Investigator Award in Science 212262/Z/18/Z to M.L.).

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

# A Additional Experiments

In this section we describe some additional experiments and again compare the performance of SOFO with Adam.

## A.1 One-shot classification

We also tested SOFO on a difficult one-shot classification task (see Appendix E.5 for details). This task relies on three Gaussian distributions of 3D inputs: $p_i(\boldsymbol{u}) = \mathcal{N}(\boldsymbol{u}; \boldsymbol{e}_i, I)$ where $I$ is the $3 \times 3$ identity matrix and $\boldsymbol{e}_i$ denotes its $i^{\text{th}}$ column. In each trial, these three input distributions are randomly assigned a unique class label $\in \{1, 2, 3\}$, and the network begins the trial with a phase of exposure to a sequence of three input/output pairs ("support set" in meta-learning jargon), each sampled from a different class and presented for $T_{\text{learn}} = 10$ consecutive time steps. In a second phase, the network is presented with three other fresh inputs from each class ("query set", $T_{\text{exploit}} = 10$), in random order, and is asked to classify them according to the same input/class contingencies (Figure 7A). With these contingencies being randomized in every trial, the RNN must use the initial exposure phase to learn from the support set, and remember and exploit this information to accurately classify the query set – and do all this using its own dynamics with fixed parameters.

SOFO outperforms Adam on this task too, successfully training an RNN to 90% one-shot classification accuracy (later 100%) in 150 training iterations, compared to 1500 iterations for Adam. Contrary to SOFO, Adam's training loss exhibits the usual plateau characteristic of first-order optimization of loss functions with pathological curvature (Figure 7B).

## A.2 Delayed addition

To demonstrate that SOFO can also train networks on tasks that require complex temporal dependencies, we apply SOFO to the delayed-addition task (Schmidt et al. [45]; see Appendix E.6 for details) with varying sequence lengths (Figure 8). There, SOFO again outperforms Adam on tasks involving both short sequences (Figure 8A) and long sequences (Figure 8B).

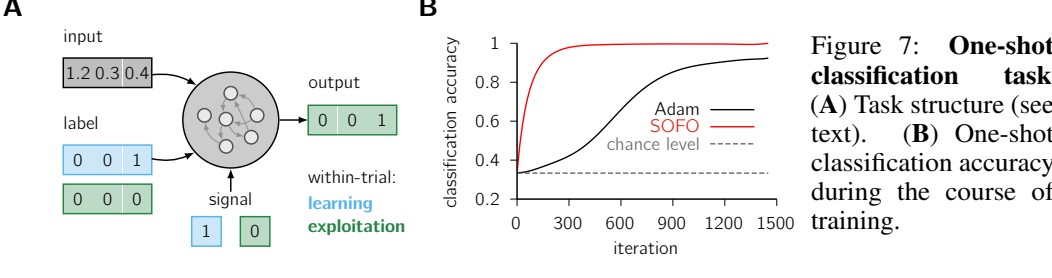

Figure 7: **One-shot classification task**. (**A**) Task structure (see text). (**B**) One-shot classification accuracy during the course of training.

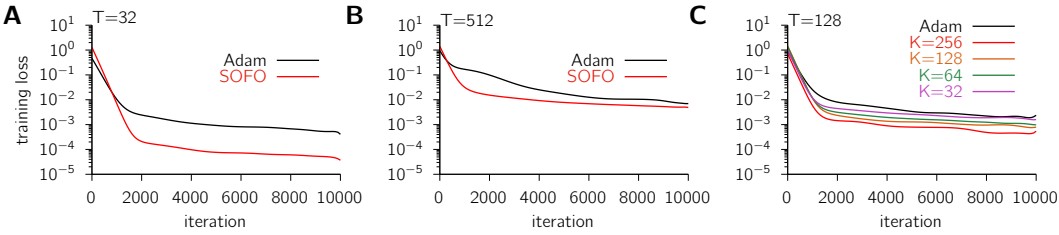

Figure 8: **Delayed-addition task**. (**A**) Training losses for a time horizon $T = 32$. (**B**) Same as (A), for $T = 512$. (**C**) Same as (A-B), for $T = 128$, but varying the number of random parameter tangents ($K$) in SOFO. In both (A–B), we had set $K = 256$.

# B Batched-FORCE

We also extend the FORCE learning algorithm [48] to its batched version, which we used for the flip-flop task in Section 4.3 and Figure 3. Recursive least squares (RLS)-based FORCE learning, as presented in [48], assumes a batch size of 1, which does not fully exploit GPU parallelism. The batched FORCE algorithm presented in Algorithm 2 implements FORCE learning on a whole batch of trials in parallel.

---

**Algorithm 2** Batched FORCE

---

1: **hyperparameters:** $\alpha$; batch size $M$; number of neurons $S$; output dimension $N$; time horizon $T$.
2: **input:** initial inverse covariance matrix $P_0 = \frac{1}{\alpha}I$, initial output weights $W_0 \in \mathbb{R}^{N \times S}$ and initial network state $z_0 \in \mathbb{R}^{S \times M}$.
3: **for** $m = 0, 1, 2, ...$ **do** $\qquad\qquad\qquad\qquad\qquad$ ▷ iterate until convergence
4: $\qquad$ sample data mini-batch with targets $\{\bar{y}_t \in \mathbb{R}^{N \times M}\}_{t=0}^{t=T}$
5: $\qquad$ **for** $t = 0, 1, 2, \ldots, T$ **do** $\qquad\qquad\qquad\qquad$ ▷ iterate over the horizon length
6: $\qquad\qquad$ obtain $z_t$ from $z_{t-1}$ and $y(t)$ (Equation 15)
7: $\qquad\qquad$ $r_t = \phi(z_t)$
8: $\qquad\qquad$ $y_t = W_t r_t$
9: $\qquad\qquad$ $e_t = y_t - \bar{y}(t)$ $\qquad\qquad\qquad\qquad\qquad\qquad$ ▷ calculate error
10: $\qquad\qquad$ $P_t = P_{t-1} - P_{t-1} r_t (M\mathbf{I}_M + r_t^T P_{t-1} r_t)^{-1} r_t^T P_{t-1}$ $\qquad$ ▷ update $P$
11: $\qquad\qquad$ $W_{t+1} = W_t - \frac{1}{M} e_t r_t^T P_t$ $\qquad\qquad\qquad\qquad\qquad$ ▷ update $W$
12: $\qquad$ **end for**
13: $\qquad$ $P_0 \leftarrow P_T$
14: $\qquad$ **output:** $\{y_t\}_{t=0}^{t=T}$
15: **end for**

---

# C Memory and runtime complexity for SOFO

This section states the memory and runtime complexity of SOFO when applied to an RNN with state dimension $S$ (assumed to be larger than the output dimension $N$), minibatch size $M$, time horizon $T$ and tangent batch size $K$.

**Memory complexity** For a recurrent neural network operating in batched forward-mode AD, what has been computed at time $t$ can be immediately discarded at time $t+1$, leading to $\mathcal{O}(T^0)$ complexity where $T$ is the time horizon. At each time step, the memory cost is dominated by the largest tangent tensor being manipulated, which will typically be the batch of tangents associated with the network's activity, of shape $S \times M \times K$. For large networks with $S > M$, the dominant contribution will be from the batch of tangents associated with the recurrent weight parameters ($\Theta$) of shape $S \times S \times K$. Note that the accumulated GGN sketch $\tilde{G}$, of size $K \times K$, makes a negligible contribution to memory usage. Thus, SOFO's memory complexity is $\mathcal{O}(\max(S, M)SK)$, to be compared to $\mathcal{O}(S^2 + SMT)$ for standard reverse-mode AD.

**Runtime complexity** SOFO's per-time-step runtime complexity is dominated by the temporal updating of the batch of activity tangents; for RNNs, this involves batched matrix multiplications which have complexity $\mathcal{O}(S^2 MK)$. Runtime complexity is also linear in the time horizon, leading to an overall $\mathcal{O}(S^2 MKT)$ complexity. Whilst this is technically $K$ times more than the $\mathcal{O}(S^2 MT)$ runtime complexity of reverse-mode AD, efficient batching of those $K$ multiplications on GPU is such that, in our experiments, SOFO was only a small constant (typically between 1.5 and 3) times slower than Adam in terms of wallclock time per training iteration (c.f. Figure 6B). Note also that the SVD of the GGN sketch (Algorithm 1, line. 10) will incur a negligible $\mathcal{O}(K^3)$ runtime cost unless $K$ is very large (which typically isn't the case anyway, because of memory constraints – see 'Memory complexity' above).

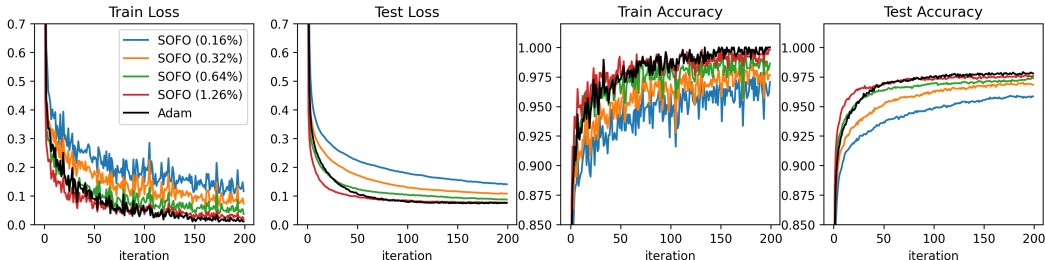

Figure 9: **MNIST classification task.** Training and test performance (see titles) for Adam (black) and SOFO (colored) with different numbers of tangents $K$. Percentages refer to the $K/P$ ratio, i.e. number of tangents $K$ to total parameter count $P$.

# D    Additional profiling experiments

Here we present additional profiling experiments on SOFO to demonstrate how SOFO performs when we (1) increase the number of tangents $K$; (2) increase the number of neurons $S$ and (3) scale up to deep networks.

## D.1    Investigating the effect of $K$ in SOFO

The choice of $K$ affects the quality of SOFO's gradient and GGN sketches – the larger, the better. In general, $K$ should be chosen to be small enough such that the training is not too memory-intensive, but large enough to ensure fast convergence. Our experiments with SOFO have shown that performance mainly depends on the ratio of $K$ to the number of parameters $P$. Strikingly, in RNN settings, this $K/P$ ratio can be chosen as low as $1\%$ and still yield excellent training performance.

Here we carry out two further experiments to demonstrate the effect of $K$ on SOFO-based training.

**MNIST classification**    We investigated the effect of $K$ on the training of a two-layer perceptron (MLP) on MNIST classification. With hidden size $100$, and input size $28 \times 28 = 784$, the network had a large total parameter count of $P = 79510$. As we increased the ratio $K/P$ from $0.16\%$ to $1.26\%$, training became gradually better, with SOFO eventually outperforming Adam in terms of progress per training iteration (Figure 9).

**Delayed addition**    We also investigated the impact of $K$ in the training of an RNN on the delayed-addition task (Appendix A.2). There, the vanilla RNN with $S = 128$ neurons had a smaller parameter count $P = 17024$ (see network structure in Appendix E.6). In this case, even using only $K = 32$ tangents ($K/P \approx 0.2\%$) is sufficient for SOFO to outperform Adam, and increasing $K$ leads to even faster convergence (Figure 8C).

## D.2    Training SOFO on large networks

In all previous experiments SOFO has mostly been employed to train small networks (i.e. up to $1000$ neurons in the flip-flop task in Section 4.3). In Figure 3D, we compared the performance of SOFO and Adam in the context of the flip-flop task, as we increased the number of neurons $S$ from $1000$ to $8000$. While increasing $S$ improved the performance of Adam in terms of convergence speed, it actually hindered SOFO. In light of the discussion in Appendix D.1, this is perhaps not surprising, as we kept the number of tangents $K$ fixed at $256$, such that the $K/P$ ratio decreased quadratically with growing $S$. In other words, with growing $S$, SOFO got to explore a diminishing fraction of the parameter space in each training iteration. Note that SOFO nevertheless outperformed Adam for all network sizes in this benchmark.

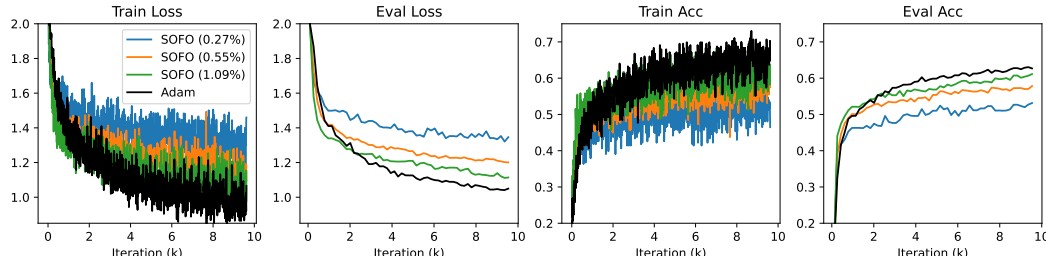

Figure 10: **Training an MLP-mixer on CIFAR-10 classification**. (**A**) Training loss. Percentages refer to the $K/P$ ratio, i.e. number of tangents $K$ to total parameter count $P$. (**B**) Evaluation loss. (**C**) Training accuracy. (**D**) Evaluation accuracy.

### D.3 Training SOFO on deep networks

We have also added a comparison between Adam and SOFO on an MLP-Mixer architecture [52] trained on CIFAR-10 classification ($\sim 10^5$ parameters; see details in Appendix E.7; Figure 10). This is a fairly easy target for Adam due to the presence of skip connections and layer-norm operations, and with the limits imposed by GPU memory, the best SOFO does here is to nearly match Adam's performance. Note that this deep network setup is not where we expect SOFO to be the most useful anyways, as this network is not deep enough to really challenge reverse-mode AD in terms of memory complexity.

## E  Details on experiments

In this section we present the details on main experiments, describing the task setup and network structure.

### E.1  Details on Figure 1

**Task structure**  The task consists of learning the dynamics of a Lorenz attractor with minimal supervision. The Lorenz attractor consists of a three-dimensional state $(x_1, x_2, x_3)$ evolving as

$$\dot{x}_1 = 10(x_2 - x_1) \quad \dot{x}_2 = x_1(28 - x_3) - x_1 \quad \dot{x}_3 = x_1 x_2 - \frac{8}{3}x_3 \tag{9}$$

Data is generated by integrating Equation 9 over with a time step of $0.01$ using a Runge-Kutta solver (RK4) followed by z-scoring. During training, the network runs for a total of $T_{\text{Lorenz}} = 32$ time steps but the only label provided is the system's state at the final time step.

**Network structure**  We use a simple RNN network with an inverted bottleneck structure for this task. The hidden state $z$ of the recurrent network, which is three-dimensional and directly used as the state readout, evolves as:

$$z_{t+1} = Az_t + W\phi(Cz_t + b) \tag{10}$$

where $A, C, W, b$ are parameters and $\phi$ is the ReLU function. The size of the inverted bottleneck is chosen to be 400 (i.e. $W$ has shape $400 \times 3$). We use the squared error (SE) between the model prediction and the label as the loss function for training of the network.

### E.2  Details on Figure 2

**Task structure**  The task consists of learning to perform adaptive Kalman filtering on a session comprising of multiple linear dynamical systems (LDSs) (i.e. the environment is non-stationary).

The state transition dynamics of an LDS is given as:

$$x_{t+1} = ax_t + v + \sigma_\epsilon \epsilon^x_{t+1} \tag{11}$$

where $x_t$ is the hidden state at timestep $t$. The state emission dynamics is given by:
$$y_t = x_t + \sigma_\beta \epsilon_t^y \tag{12}$$
where $y_t$ is the observation at timestep $t$ (Figure 2A). $a$ is the state transition parameter, $b$ is the stationary mean of the process and $v$ is the drift at each time point and is obtained as $v = (1-a)b$. $\sigma_\epsilon^2$ is the process noise variance and $\sigma_\beta^2$ is the observation noise variance. $\epsilon^x$ and $\epsilon^y$ are both drawn i.i.d. from a standard Gaussian distribution $\mathcal{N}(0,1)$. In particular, $\sigma_\epsilon^2$ is related to the stationary variance of the LDS, which we denote $\sigma^2$, via $\sigma_\epsilon^2 = (1 - a^2)\sigma^2$.

Each session has a horizon length of $T$ timesteps and consists of a number of LDSs: an LDS survives for the first $T_{\text{fix}}$ number of timesteps, and from $T_{\text{fix}}$ beyond at every step there is a $1/d$ probability of transitioning onto a new LDS, where $d$ is denoted as the survival time. Before the start of each LDS, we first sample its parameters (i.e. $a$, $b$, $\sigma_\beta^2$) from a common global distribution, before sampling the first data point for this particular LDS from $\mathcal{N}(b, \sigma^2)$. Hence, each session is characterized by a number of LDSs whose transitions are iid. The values of parameters and the global distributions from which the parameters are sampled are presented in Table 1.

Given the data emitted by the LDSs as described above, the task is as follows: at each time step $t$, the neural network receives a new observation $y_t$, and outputs a prediction of the current latent state $\hat{x}_t$. The training loss is a simple temporal accumulation of squared prediction errors. The network is hence trained to amortize inference (approximating $p(x_t|y_{1:t})$) in this non-stationary Kalman filter setting.

**Network structure**  We use a vanilla RNN with for this task. The momentary hidden state $z_t$ of the recurrent network evolves as:
$$z_{t+1} = (1-\alpha)z_t + \alpha(\phi(Cz_t + By_t + e)) \tag{13}$$
where $B, C, e$ are network parameters and $\phi$ is the ReLU nonlinearity. The estimated current state is obtained as:
$$\hat{x}_t = Oz_t \tag{14}$$
where $O$ is the readout matrix. The number of neurons is 10 (dimension of $z$) and $\alpha = 0.25$. Network parameters are initialized so as to reproduce the current observation (i.e. $\hat{x}_1 = y_1$). We use the squared error between the predicted current state $\hat{x}_t$ and the actual current state $x_t$ averaged over $T$ timesteps as the loss function.

**Kalman filter and Kalman random baselines**  The 'Kalman filter' baseline has access to the parameters of each LDS (i.e. $a, b, \sigma^2, \sigma_\beta^2$) and is hence aware of each context change. The 'Kalman random' baseline performs the same updates as the Kalman filter, except for that it has knowledge of $b$ but not of the other parameters ($a, \sigma^2, \sigma_\beta^2$). Instead, it assumes random values for these parameters, drawn from the same global distribution as the one used to generate the parameters for each LDS (Table 1).

### E.3   Details on Figure 3

**Task structure**  We conduct a 3-bit flip-flop task as in Sussillo and Barak [49], where three input channels sporadically provide bits ($\pm 1$) at random time intervals while remaining at zero at other

Table 1: Values of LDS parameters and global distributions for the sampling of LDS parameters in learning an adaptive Kalman filter task (Section 4.2) $p(x)$ denotes the distribution that $x$ is drawn from and $\mathcal{U}(a, b)$ denotes the uniform distribution between $a$ and $b$.

| parameters | distribution/values |
|---|:---:|
| $T$ | 5000 |
| $T_{\text{fix}}$ | 100 |
| $d$ | 100 |
| $p(b)$ | $\mathcal{U}(-3, 3)$ |
| $p(\tau)$ | $\mathcal{U}(1, 10)$ |
| $p(\sigma_\beta)$ | $\mathcal{U}(0.1, 3)$ |
| $p(\sigma)$ | 1 |

times. Each corresponding output channel must sustain its activity ($\pm 1$) at the value given by the last provided input bit. In other words, the network must maintain at least $2^3 = 8$ bit memory at any time instance, while ignoring the cross-talk between differing input/output pairs.

**Network structure** The hidden state $\boldsymbol{z}$ of the recurrent network evolves as:

$$\boldsymbol{z}_{t+1} = (1 - \alpha)\boldsymbol{z}_t + \alpha(\phi(J\boldsymbol{z}_t + B\boldsymbol{y}_t + W_{\text{FB}}\hat{\boldsymbol{x}}_t)) \tag{15}$$

where $J, B, W_{\text{FB}}$ are network parameters, $\boldsymbol{y}$ is the output and $\phi$ is the ReLU nonlinearity. The output is obtained as:

$$\hat{\boldsymbol{x}}_t = W\phi(\boldsymbol{z}_t) \tag{16}$$

where $W$ is the readout matrix. Note that to similar to FORCE learning, we fix all weights and only train output weights $W$. In particular, entries of $J$ are drawn from a Gaussian distribution $\mathcal{N}(0, \frac{g^2}{S})$, where $g > 1$ initializes the network in the chaotic regime (which is necessary for FORCE learning) but $g < 1$ initializes the network in the non-chaotic regime. We use $S = 1000$ neurons and mean squared error between the target trace and output trace as the training criterion.

### E.4 Details on Figure 4 and Figure 5

**Task structure** See main text for an overall description of the task. The details of the two-jointed arm model used in these tasks can be found in [23]. The central spot is about 20cm in front of the shoulder joint. For the single-reach task, the duration of the idle phase is drawn uniformly between 0.1 and 0.5s in each trial, and the duration of the preparation phase is drawn uniformly between 0.2 and 0.8s. Each trial lasts for 2 seconds regardless of whether the target has been successfully reached. During the idle & preparation epochs, the loss function is a quadratic penalty on the squared deviation of the hand position from the central spot and a quadratic penalty on the hand velocity. During the movement epoch, there is no loss for the first 600ms, and then the deviation of the hand from the desired reach target is penalized quadratically, together with the hand velocity.

For the double-reach task, each trial begins with the preparation phase right away, whose duration is sampled uniformly between 0.2 and 0.6s. The loss is similar to that described above, with one more term for the second target to be reached.

For the single-reach task, the input $\boldsymbol{u}(t)$ is 3-dimensional, the first two channels relaying target information and the third conveying the go cue [50]. For the double-reach task, the input is 5-dimensional, as two more dimensions are needed for the second target.

**Network structure** The stochastic vanilla network dynamics considered here read:

$$\tau\frac{d\boldsymbol{z}}{dt} = -\boldsymbol{z} + W(\phi(\boldsymbol{z}) \odot (1 + \sigma\boldsymbol{\epsilon})) + \boldsymbol{b} + B\boldsymbol{u} + F(\boldsymbol{\theta}_1, \boldsymbol{\theta}_2, \dot{\boldsymbol{\theta}}_1, \dot{\boldsymbol{\theta}}_2)^\top \tag{17}$$

where $(\boldsymbol{\theta}_1, \boldsymbol{\theta}_2)$ and their derivatives denote the Markov state of the two arm joint angles. Here, $W$ is a matrix of recurrent weights, $\boldsymbol{b}$ is a bias vector, $B$ is the input weight matrix, and $F$ is a matrix of feedback weights. We set $\tau = 20$ ms, and use networks with 64 neurons. The nonlinearity $\phi(\cdot)$ is the ReLu function, and "firing rates" are corrupted by multiplicative noise $(1 + \sigma\boldsymbol{\epsilon})$ where $\boldsymbol{\epsilon}$ is an $N$-dimensional random vector of uniformly sampled elements between $-0.5$ and $0.5$.

The network dynamics are integrated using the Euler method with a time step of 2 ms.

### E.5 Details on Figure 7

**Task structure** The task is a one-shot three-class classification. We define three Gaussian distributions of three-dimensional (3D) inputs: $p_i(\boldsymbol{u}) = \mathcal{N}(\boldsymbol{u}; \boldsymbol{e}_i, I)$ where $I$ is the $3 \times 3$ identity matrix and $\boldsymbol{e}_i$ denotes its $i^{\text{th}}$ column. In each trial, these three input distributions are randomly assigned a unique class label $\in \{1, 2, 3\}$. Each trial consists of two phases: the learning phase and the exploitation phase. During the learning phase, the network receives a set of three components (the support set): the input is a randomly generated example, the label is the corresponding one-hot encoded class label and a teaching signal of value 1. The same set is presented for a fixed $T_{\text{learn}} = 10$ timesteps before the next set is presented for another $T_{\text{learn}}$ timesteps where the input is generated from a different class, until all three classes have been presented. During exploitation the network receives a set of two components (the query set): the input is a randomly generated example and a teaching signal of value

0 (all entries in the label is set to zero). Similar to the learning phase, the input is an unseen example drawn from one class is presented for a fixed $T_{\text{exploit}} = 10$ timesteps, before another example drawn from a different class is presented for another $T_{\text{exploit}}$ timesteps, until examples from all three classes have been presented. The goal of the neural network is to predict the class at the last step of $T_{\text{exploit}}$ for each of the 3 unseen examples during the exploitation phase (Figure 7A).

**Network structure**  We use a simple RNN with an inverted bottleneck for this task. The hidden state $z$ of the recurrent network evolves as:

$$\boldsymbol{z}_{t+1} = A\boldsymbol{z}_t + W\phi(C\boldsymbol{z}_t + \boldsymbol{e}) + B\boldsymbol{y}_t + \boldsymbol{d} \tag{18}$$

where $A, B, C, W, \boldsymbol{e}, \boldsymbol{d}$ are network parameters, $\phi$ is the ReLU nonlinearity and $\boldsymbol{y}_t$ is the input and teaching signal concatenated together at the time step $t$. The predicted next step is obtained as:

$$\hat{\boldsymbol{y}}_t = O\boldsymbol{z}_t \tag{19}$$

where $O$ is the readout matrix. The number of neurons is $100$ (dimension of $\boldsymbol{z}$) and the size of the inverted bottleneck is $400$ ($W$ is of size $400 \times 10$). We use the cross-entropy between the true label and the network output as the loss function, averaged over the $c = 3$ predictions that the network makes during each session.

### E.6  Details on Figure 8

We conduct a delayed addition task as in Schmidt et al. [45]. The input consists of two sequences both of length $T$: the first sequence contains entries drawn from a uniform random distribution $\mathbf{U}[0, 1]$ while the second sequence contains zeros except for one at two random timings: $t_1 < 10$ and $t_2 < T/2$. Constraints on $t_1$ and $t_2$ are chosen such that every trial requires a long memory of at least $T/2$ time steps. At the last time step $T$, the network is expected to generate an output that is the sum of the two numbers in the first sequence at $t_1$ and $t_2$.

We use a vanilla RNN (same structure as that in Appendix E.2) with $S = 128$ neurons to learn the task. We use the mean squared error between the target and output as the training criterion.

### E.7  Details on Figure 10

We use SOFO to train a five-layer MLP-mixer [52] on CIFAR-10 classification task. The network consists of a patchification layer (patch size $4 \times 4$) followed by a per-patch linear embedding to a higher channel dimension of $128$. This is then followed by two mixer blocks, each block including one token-mixing MLP with dimension $64$, and one channel-mixing MLP with dimension $128$ (each MLP uses a GELU nonlinearity). Finally, a fully-connected output layer performs classification. Other components such as skip-connections, dropout and layer norm on the channels are also included. This adds up to about $P \approx 10^5$ parameters.

## F  Hyperparameter values

Here we present the ranges of hyperparameter values tested for Adam and SOFO in Table 3 and Table 5 respectively all experiments carried out. The final values of hyperparameters used in the experiments for Adam and SOFO are presented in Table 2 and Table 4 respectively. We finally present the range of hyperparameters tested and the final hyperparameters for FGD and first-order SOFO for the Lorenz task (Section 4.1) and for batched FORCE in the 3-bit flipflop task (Section 4.3) in Table 6.

Table 2: Hyperparameters used for Adam. $\eta$ is to the learning rate.

|  | batch size | $\eta$ |
|---|---|---|
| **Lorenz** | 256 | $2 \times 10^{-4}$ |
| **Kalman** | 256 | $10^{-2}$ |
| **one-shot** | 256 | $10^{-5}$ |
| **single reach** | 256 | $10^{-3}$ |
| **double reach** | 256 | $10^{-3}$ |
| **flip-flop** | 32 | $10^{-4}$ |
| **delayed addition** | 256 | $10^{-4}$ |
| **MNIST** | 512 | $9 \times 10^{-4}$ |
| **cifar-10** | 256 | $2 \times 10^{-4}$ |

Table 3: Ranges of hyperparameters explored for Adam

|  | batch size | $\eta$ |
|---|---|---|
| **Lorenz** | 256 | $10^{-4} - 10^{-2}$ |
| **Kalman** | 256 | $10^{-4} - 10^{-2}$ |
| **one-shot** | 256 | $10^{-5} - 10^{-3}$ |
| **single reach** | 256 | $10^{-6} - 10^{-5}$ |
| **double reach** | 256 | $10^{-6} - 10^{-4}$ |
| **flip-flop** | $32 - 256$ | $10^{-5} - 10^{-2}$ |
| **delayed addition** | 256 | $10^{-5} - 10^{-2}$ |
| **MNIST** | 512 | $10^{-4} - 10^{-2}$ |
| **cifar-10** | 256 | $10^{-4} - 10^{-3}$ |

Table 4: Hyperparameters used for SOFO. $K$ is the number of tangents used, $P$ is the total number of network parameters, $\eta$ is the learning rate and $\lambda$ is the relative damping applied on SOFO. Note that for experiments where a range of $K$ is used the corresponding entry in the table is left as empty.

|  | batch size | $K$ | $K/P$ ratio | $\eta$ | $\lambda$ |
|---|---|---|---|---|---|
| **Lorenz** | 256 | 128 | 4.6% | 0.3 | $10^{-5}$ |
| **Kalman** | 256 | 256 | 2.5% | 0.05 | $10^{-3}$ |
| **one-shot** | 256 | 256 | 2.8% | 0.1 | $10^{-3}$ |
| **single reach** | 256 | 256 | 5.4% | 0.1 | $10^{-6}$ |
| **double reach** | 256 | 256 | 5.2% | 0.1 | $10^{-6}$ |
| **flip-flop** | 32 | 256 | 2.5% | 0.1 | $10^{-5}$ |
| **delayed addition** | 256 | - | - | 0.1 | $10^{-5}$ |
| **MNIST** | 512 | - | - | 0.8 | $10^{-7}$ |
| **cifar-10** | 256 | - | - | 0.5 | $10^{-7}$ |

Table 5: Ranges of hyperparameters explored for SOFO

|  | batch size | $K$ | $\eta$ | $\lambda$ |
|---|---|---|---|---|
| **Lorenz** | 256 | 128 | $10^{-2} - 0.7$ | $10^{-7} - 10^{-3}$ |
| **Kalman** | 256 | 256 | $0.5 - 1$ | $10^{-5} - 10^{-3}$ |
| **one-shot** | 256 | 256 | $10^{-2} - 0.2$ | $10^{-4} - 10^{-2}$ |
| **single reach** | 256 | 256 | $10^{-3} - 1$ | $10^{-7} - 10^{-4}$ |
| **double reach** | 256 | 256 | $10^{-3} - 1$ | $10^{-7} - 10^{-4}$ |
| **flip-flop** | 32-256 | 256 | $10^{-2} - 1$ | $10^{-7} - 10^{-3}$ |
| **delayed addition** | 256 | - | $10^{-4} - 0.1$ | $10^{-7}$ |
| **MNIST** | 512 | - | $0.1 - 1$ | $10^{-5} - 10^{-4}$ |
| **cifar-10** | 256 | - | $0.1 - 1$ | $10^{-6}$ |

Table 6: Range of the learning rate $\eta$ explored and all final hyperparameters used for FGD [4], first-order SOFO [28] for the Lorenz task (Section 4.1) and batched FORCE (Appendix B) for the 3-bit flip-flop task (Section 4.3). Range of $a$ refers to the range of the parameter $a$ explored.

|  | batch size | $K$ | $\eta$ | range of $\eta$ | $\alpha$ |
|---|---|---|---|---|---|
| **FGD** | 256 | 1 | $2 \times 10^{-4}$ | $10^{-6} - 10^{-4}$ | - |
| **first-order SOFO** | 256 | 128 | $10^{-2}$ | $10^{-6} - 10^{-4}$ | - |
| **batched FORCE** | 256 | - | - | - | $0.1 - 100$ |

