# OpenReview forum: "Second-order forward-mode optimization of recurrent neural networks for neuroscience"
_NeurIPS.cc/2024/Conference — NeurIPS 2024 spotlight_

### Official Review · Reviewer_3Uai · 2024-06-28

**Soundness:** 3
**Presentation:** 4
**Contribution:** 4
**Rating:** 8
**Confidence:** 4

**Summary:**

The present work addresses the difficulty to optimize RNNs that are used in the realm of neurosciences to model biological data, which by design may not benefit from modern RNN inductive biases (e.g. gates), nor be trainable by the Adam optimizer and may inherently hit memory limits due to handling long temporal sequences. To this end, the paper proposes a forward-only second-order optimizer which computes and invert a Gaussian-Newton curvature matrix in a random parameter subspace spanned by $K$ randomly sampled directions in the parameter space. The resulting algorithm amounts, up to a reparametrization of the loss, to exact Gauss-Newton optimization within this parameter subspace. Algorithmically, the procedure amounts to simultaneously computing activations and directional derivatives, a.k.a Jacobian-vector products (JVPs), throughout the forward computational path, for each of the K random directions. As such, the algorithm doesn't require storing any activations and can be parallelized across the K random directions on different GPUs. The algorithm is largely superior to Adam on a variety of neuroscience-oriented tasks in terms of training speed, resulting performance and memory footprint.

More precisely:
- Section 2 formulates the optimization problem, introduces standard generalized Gauss-Newton optimization and forward-mode automatic differentiation and briefly surveys subspace optimization methods.
- Section 3 introduces SOFO (Second Order Forward Only optimizer) reading as follows: at each training iteration, sample K random directions $u_k$ in the parameter space, and compute directional derivatives for the model's Jacobian $Y_k := \partial_\theta y \cdot u_k$ and cost function $C_k := \partial_\theta c^\top \cdot u_k$ for each of these. Build the GGN matrix within the random subspace with entries $G_k := Y_k^\top \cdot \partial_y^2 L \cdot Y_k $, compute GN update $\Delta\tilde{\theta}$ in the random subspace coordinates, and project it back to the original parameter space $\Delta \theta := \sum_{k=1}^K \Delta\tilde{\theta}_k \theta_k$.
- Section 4 presents results on four different tasks:
  i) an RNN learns the trajectory of a 3-dimensional Lorentz attractor with sparse error signal (Section 4.1,). SOFO significantly outperforms Adam, its first-order counterpart (i.e. without gradient preconditioning) using $K$ directions, and a single direction ("FGD").
  ii) Adaptive Kalman filtering where observations are produced from ground-truth latents (Linear Dynamical Systems), and an RNN is made to learn to recover these latents from observations. The ground-truth parameters of the LDS may change sporadically, yielding successive such LDS's, making the task more difficult for the RNN to learn. Still here, SOFO outperforms Adam and two baselines where the ground-truth LDS's parameters are known.
 iii)  SOFO is also evaluated on one-shot classification where an RNN is presented with samples from a support set with one example per class (each of these classes being a 3D Gaussian distribution) and has to classify three other samples from a query set. Here again, SOFO is superior to Adam.
iv) Finally, SOFO is evaluated on motor tasks where a target has to be memorized ("idle" and "preparation" phase) and reached ("movement" phase), either a single one ("single reach task") or two of these and in a sequential fashion ("double-reach" task). The RNN learns to output the appropriate torques to move the arm towards the targets. It is shown that while Adam struggles on the single reach task and totally fails on the double-reach task, SOTA successfully learns these two tasks in terms of resulting performance and training speed, although SOFO runtime exceeds that of Adam. Arm trajectories are depicted to illustrate and compare SOFO and Adam easily on the two tasks, and it is shown that neurons specialize in one of the three phases.
- Finally, it is shown that while the memory footprint of Adam scales linearly with the sequence length of the task at hand, the memory cost of SOFO is (as expected) constant.
- It is also worth mentioning that SOFO is tested on larger models on MNIST where it performs on par with Adam (Appendix, Fig. 7).

**Strengths:**

- All the claims made about SOFO hold: it significantly outperforms Adam on all tasks, it has a constant memory footprint because forward-only, it amounts to exact Gauss-Newton optimization in the random parameter subspace.
- The paper is extremely neat and clear: the structure and language of the paper, the mathematical notations, the SOFO pseudo-algorithm, the presentation of the results.
- The figure quality is high.
- 4 different experimental setups are investigated. Beyond showing the performance of SOFO, I think it was not trivial to find a set of tasks where Adam fails and which is also of relevance to neuroscience.

**Weaknesses:**

- The models at use are very small, and at most two-layer deep in space (i.e. each hidden states at a given timestep can be inferred with at most two matrix multiplications).
- I think that the proposed method should be properly benchmarked against RTRL (which is also an **online**/**forward-only** technique, constant in memory) and more complex temporal tasks (see questions below).
- I'm not sure that the memory profiling analysis is the most relevant here. See my question below.
- Architectural details about the MNIST experiment in the Appendix are not given.

**Questions:**

- For memory profiling: I think that a forward-only equivalent of BPTT, namely Recurrent Real Time Learning (RTRL), should instead be considered here in combination with Adam. Just to clarify: using RTRL in place of BPTT will not yield any difference in performance in all the tasks considered as these are equivalent, so I am not questioning the relevance of the results. However, using RTRL instead of BPTT has a different memory cost. Generally, RTRL is discarded because of the cubic memory cost of storing $\partial_\theta h$ across time steps. However, because the dimension at use in all your experiments are pretty small, I don't think the memory cost of RTRL would be that huge. **Could you please carry out the memory profiling using RTRL**?
- Could you please provide architectural details of the MNIST experiments?
- Did you try evaluating SOFO on CIFAR-10? If you could evaluate SOFO on a 6/7 layers-deep architecture on CIFAR-10, that would be a great cherry on the cake.
- L.1 "a common source of anxiety", L.116 "embarrassingly parallelizable", L.165 "embarrasingly parallel problem": why be so embarrassed and anxious? I don't think that this language adds much value to the already great presented work.
- **Could you consider one of these more complex temporal tasks to check your method against**: sCIFAR, IMDB, ListOps? Taking these from "Online learning of long range dependencies" [Zucchet et al, 2023] where RTRL is used as an online learning method which is also constant in memory.

**Limitations:**

See my questions above. I would be very happy to have extra experiments by the end of the discussion period.

---

> ### Author Rebuttal · Authors · 2024-08-07
>
> We thank you for your thoughtful review (and for the extensive summary of our paper!). In the main rebuttal above, we have addressed the concerns common to all reviews, including your comments on small network sizes. Here, we wish to address your more specific concerns.
>
> Thank you for bringing up RTRL, which also has $\mathcal{O}(T^0)$ memory cost. Having thought about it, RTRL is mathematically equivalent to a particular limit of SOFO: the limit where (i) SOFO doesn't use curvature information (i.e. dropping the inverse matrix term in Eq. 8) and (ii) the set of tangent vectors used at each iteration ($\Theta$ in Eqs. 6 and 7) is a full basis equal to the identity matrix. In this limit, batched forward-mode differentiation (efficiently) implements the usual RTRL recursion to propagate exact, entire Jacobians of network activity w.r.t. the parameters, in (constant memory) forward mode. We can therefore predict that, for a model with $P$ parameters, the memory cost of RTRL will be $P/K$ times that of SOFO if SOFO uses $K$ random tangents. Given that the ratio $K/P$ used in our experiments is around $0.01$, this would make RTRL ~100 times more memory-hungry than SOFO. This means that unless we compensate by drastically reducing the (data) batch size, we wouldn't be able to run RTRL on our tasks with our architectures. To confirm these predictions, we implemented RTRL in our codebase on the Kalman example of Figure 2, and will be adding these new data points to the memory profiling of Figure 6. To summarize our findings: RTRL forces us to reduce the batch size to less than 10, and even then it nearly fills our GPU memory (12GB). In contrast, SOFO can easily accommodate a batch size of 256, at which point it only fills 1GB of memory. Thanks very much for reminding us of RTRL, it is in fact rather nice that we are able to frame RTRL as a particular limit of SOFO; we will add a paragraph of discussion on this.
>
> Regarding running more complex tasks: we haven't had a chance to run any of the tasks you suggested from Zucchet et al., 2023. Instead, we followed Rev. Z9Qd's suggestion of adding a delayed addition task, with results in Figure 2 of the one-page rebuttal PDF. In this task, SOFO again outperforms Adam.
>
> We also worked to provide the “great cherry on the cake”, by running SOFO on a 5-layer MLP-Mixer (Tolstikhin et al., 2021). This network consists of a patchification layer (patch size $(4,4)$) followed by a per-patch linear embedding to a higher channel dimension of $128$. This is then followed by two mixer blocks, each block including one token-mixing MLP with dimension $64$, and one channel-mixing MLP with dimension $128$ (each MLP uses a GELU nonlinearity). Finally, a fully-connected output layer performs classification. Other components such as skip-connections, dropout and layer norm on the channels are also included. This adds up to about $P \approx 10^5$ parameters. We compared SOFO with Adam under increasing $K/P$ ratio (with $K$ the number of tangents), from $0.27$% to $1.09$%, with results shown in Figure 3 of the one-page rebuttal PDF. As expected, SOFO gets better as this ratio increases; for $K/P = 1.09$%, SOFO approaches the performance of Adam -- so it is a cherry on a cake, just not a very sweet one (?). We sort of expected this, as this network is a fairly easy optimization target for Adam; we suspect that if we did away with the skip connections and layer-norm tricks, Adam (but not SOFO) would struggle more. We are currently exploring this and will be adding a comment later on, as results become available. Note that we have tried our best to do hyperparameter tuning for all algorithms given the short turnaround time of this rebuttal.
>
> > Could you please provide architectural details of the MNIST experiments?
>
> These can be found in L.838 in the appendix (2-layer fully-connected MLP, hidden size 100).
>
> > why be so embarrassed and anxious?
>
> (“embarassingly parallel” is a standard term in the field of parallel computing, but we are happy to replace it with the equivalent term "perfectly parallel" -- simply meaning that a parallel implementation does not require any clever strategy, the problem being inherently parallel already with no sharing of computation between instances (between the different JVPs, in our case). In any case, we agree that we could well do away with the "anxiety" vocabulary :) -- we actually had afterthoughts after submission, thinking that this phrasing was not particular thoughtful to grad students who actually struggle with serious life anxiety, as many do.)

---

> > ### Comment · Reviewer_3Uai · 2024-08-08
> > **Answer to rebuttal**
> >
> > Dear authors,
> >
> > I do appreciate a lot the great amount of work you produced for this rebuttal, thank you very much.
> > - Thank you for making the connection to RTRL clearer.
> >    + **Will you please mention this connection in the main, and add this connection in greater details somewhere in the appendix?**
> >    + Can you please update the memory profiling figure accordingly?
> >
> > - Especially, thank you for the MLP mixer experiment on CIFAR-10! This is very promising. One quick question about Fig. 3: why do we observe this very oscillatory / noisy behavior on the training quantities and not the validation ones? Is this simply because loss and accuracy are evaluated *batch-wise* for training and *over the whole validation set* for the evaluation?
> >
> > - I ignored that "embarrassingly parallel" was a common term in the field of parallel computing. Please ignore my (embarrassing) comment about this, no need to make any change if this is also standard in the eyes of the other reviewers.
> >
> > - Yes, no need at all for the anxiety vocabulary.
> >
> > All things considered and if the authors commit to the above changes, I'm happy to raise my score to strong accept!

---

> > > ### Author Response · Authors · 2024-08-08
> > >
> > > Many thanks for your enthusiastic feedback!
> > >
> > > We will make the connection to RTRL more detailed/formal in an appendix and update Figure 6.
> > >
> > > Regarding Figure 3 in the one-page rebuttal PDF: yes indeed, it is showing training losses evaluated on individual data minibatches (n=256; hence why they are so noisy), whereas eval losses are computed for the whole eval set (n=5120).

---

### Official Review · Reviewer_Z9Qd · 2024-07-10

**Soundness:** 3
**Presentation:** 3
**Contribution:** 4
**Rating:** 8
**Confidence:** 4

**Summary:**

The manuscript introduces a second order optimization algorithm (called SOFO) for training recurrent neural networks on neuroscience tasks. SOFO does not do backpropagation, instead uses a batched forward-mode differentiation that is memory efficient for large (in time) computation graphs. Authors show that RNNs trained with SOFO outperforms those trained with Adam, a common first-order optimizer for training RNNs, on various neuroscience related tasks. In these tests, SOFO is more memory efficient and converges faster.

**Strengths:**

- Being able to train RNNs on neuroscience tasks in a scalable and timely manner is an urgent need in the field.
- The fact that authors achieved these results with a relatively cheap GPU is impressive, and a welcome addition for being able to train RNNs on cheap hardware.

**Weaknesses:**

- I believe an important baseline, FORCE [1], is missing. The authors should show whether SOFO is preferable to FORCE when training RNNs on neuroscience tasks.
- I think there needs to be more concrete theoretical and empirical discussions of scaling with respect to the number of neurons.

**Questions:**

Overall, I believe the work is barely below the acceptance criteria. Yet, my score can *significantly* increase if the authors address the relevant baselines with appropriate experiments (vs FORCE [1], say on a 3-bit flip flop task as in [2] and varying numbers of neurons), and perform additional experiments to fully flesh out the extent to which SOFO is superior (scaling experiments). Please see my major questions/concerns below:

- I am sure this comment is no surprise to the authors, but the common issue with 2nd order optimizers is the inability to scale to large number of neurons. This, on the other hand, is very important for a computational neuroscience student trying to train their task. Could you please add experiments that show the scaling of SOFO w.r.t. Adam as the number of neurons increase? For example, can you show what happens once we get to the levels of ~10,000 neurons? Can SOFO train well without any memory issues?

- Relatedly, in this work, the authors are mainly using networks with only O(10) neurons. This information can only be found in the appendix. It would be helpful to have this information in relevant figure captions.

- There are several claims about the runtime complexity being comparable to a first order method, but I do not see any formal proof that includes the number of neurons as well? Could you provide the runtime complexity in terms of N and T? I see that there is an SVD computation, so would it be fair to say the complexity scales as O(K^3)? Then, how does K scale with the number of neurons N? It seems K scales linearly with N (from the appendix). Then, SOFO would not have the same complexity as gradient descent. Could the authors clarify?

- I believe the authors are missing some key citations and baselines. I would imagine FORCE [1] would be a directly relevant approach for a fast training of RNNs on neuroscience tasks, which is also a 2nd order approach and can handle relatively large number of neurons (1000 neurons in [2]). I believe the benchmarks should include comparisons with respect to FORCE whenever applicable.

- There are also some other relevant citations that may deserve some mentions, e.g. full-FORCE [3]. There are some more recent works, which the authors may want to include in their relevant work section. In general, computational neuroscience literature is no stranger to 2nd order optimizers, which I think can be emphasized a bit better in the relevant work section to place this work into the broader literature.

- The problem formulation seems to suggest that the loss function is defined over all time steps. What if I had a loss function that penalized only the output after few time steps and had no penalty during the delay period? Would SOFO also work in this case with equal efficiency? Could you show an example task if yes, or state this if no? For example, you can consider a delayed addition task [4], where the output only at the end of the trial matters.

Minor points:

- In Eq. (1), $l_{mi}(y_{mi}(.))$ seems to have only one argument. But later, it is assumed to be strongly convex in its first argument? I guess the authors refer to $\theta$, whereas $x(\theta)$ depends on $\theta$. This is also why the full function is not convex w.r.t. $\theta$? can you explicitly state this? Overall, the notation needs to be cleaned here.

- "Whilst using large batches may enable faster learning with larger learning rates, it is often the case that increasing the batch size beyond a certain point no longer helps in this respect" This sentence requires a citation.

- Lines 124-125: SOFO is not yet introduced, so these lines are confusing for the reader.

**Edit:** The revisions changed my mind and I believe this work is a significant contribution to the neuroscience literature. The results presented were beyond what I thought they would be. I support the acceptance of this work.

Citations

[1] Sussillo, D., & Abbott, L. F. (2009). Generating coherent patterns of activity from chaotic neural networks. Neuron, 63(4), 544-557.

[2] Sussillo, D., & Barak, O. (2013). Opening the black box: low-dimensional dynamics in high-dimensional recurrent neural networks. Neural computation, 25(3), 626-649.

[3] DePasquale, B., Cueva, C. J., Rajan, K., Escola, G. S., & Abbott, L. F. (2018). full-FORCE: A target-based method for training recurrent networks. PloS one, 13(2), e0191527.

[4] Schmidt, D., Koppe, G., Monfared, Z., Beutelspacher, M., & Durstewitz, D. (2019). Identifying nonlinear dynamical systems with multiple time scales and long-range dependencies. arXiv preprint arXiv:1910.03471.

**Limitations:**

I believe there needs to be a more detailed discussion of SOFO's scaling with respect to number of neurons. I think even if SOFO does not scale well, the paper could easily meet the bar for publication as long as it is stated in the main text with a proper experiment.

---

> ### Author Rebuttal · Authors · 2024-08-07
>
> Many thanks for your thoughtful review; we have been greatly encouraged this last week by your expressed willingness to potentially increase your score. In the main rebuttal above, we have addressed the concerns common to all reviews, including the effect of network size. Here, we address your other concerns.
>
> First, we appreciate your suggestion of using FORCE as another baseline. We have done that in the context of a 3-bit flip-flop task (as you suggested), with results shown in Figure 1 of the one-page PDF. We expected FORCE to be a hard baseline to beat, as RLS is in some sense the optimal thing to do in the “reservoir” setting. However, whilst FORCE does make more rapid initial progress, SOFO eventually wins. To make the comparison fair to FORCE, we implemented a batched version of FORCE-RLS (which we couldn't find anywhere, from a quick scan of the FORCE literature), using the rank-M version of the Woodbury identity for online inverse cov. estimation. This greatly accelerates FORCE learning in our experiments. Note, however, that we have only had the time to do crude tuning of FORCE's damping parameter $\alpha$ (we tried 1, 10 and 100, and 10 was the best).
>
> Interestingly, we find that SOFO actually works best when the reservoir is non-chaotic at initialization ($g=0.5$ instead of $1.5$)  -- whereas FORCE tends to fail in this setting (as is well known). What exactly that implies for neuroscience applications is unclear. Training from an initially chaotic network might result in residual chaos in corners of state space that are not relevant to the task -- and whether that is a good bias to have presumably depends on whether the brain is chaotic outside task manifolds, something on which we believe the jury is still out. At any rate, by allowing efficient training from non-chaotic initializations, SOFO stand as nicely complementary to FORCE in the RNN training toolbox.
>
> Moreover, SOFO is more flexible than FORCE: it is a general purpose 2nd-order optimizer that can be used to tweak any of the network's parameters, not only the output weights. The Full-FORCE paper nicely demonstrated that by restricting the locus of learning to those few task-relevant output weights, FORCE comes up with solutions that can be brittle. Full-FORCE solved this problem by considering more output units than strictly required by the task and constructing useful pseudo-targets for these additional outputs. This yields higher-rank weight modifications and more robust dynamics. Going further, SOFO can be used to train the _entire_ set of recurrent weights, biases, feedback and input weights. We tried this: SOFO was able to train a much smaller network of 128 units and ReLu activations (instead of the usual FORCE tanh) on the 3-bit flip flop task, to a final test MSE more than 15 times smaller than FORCE's with 1000 neurons. We find these comparisons to FORCE enlightening and will make sure to include them in the main text, not just in the appendix -- thanks again for the suggestion.
>
> Regarding runtime complexity: SOFO applied to an RNN of size $S$ over a time horizon $T$ with batch size $M$ has runtime complexity $\mathcal{O}(KTMS^2)$, plus $\mathcal{O}(K^3)$ for the SVD of the sketched GGN as you note (the latter is typically negligible relative to the former, with K being just a few hundred). In contrast, standard backpropagation through time has complexity $\mathcal{O}(TMS^2)$. Therefore, SOFO incurs an extra factor of $K$, and since K is a hyperparameter that the user is free (a priori) to set independently of $S$, this still makes SOFO technically $\mathcal{O}(S^2)$. Perhaps the question is (as you rightly point out) whether maintaining a set level of performance implicitly ties $K$ to $S$. In theory, $K$ ought to be on the same order as the effective rank of the GGN matrix (which is hard to estimate a priori). From experimenting with SOFO, we find that performance mainly depends on the ratio of $K$ to the number of parameters to be optimized, and that will be application dependent (e.g. we may only optimize low-rank modifications of the weights, or all recurrent weights, etc). Moreover, all our experiments show that SOFO works well even when this ratio is as low as ~1%. It is also worth noting that SOFO is readily parallelizable over the $K$ tangents: we find that whether we set $K=1$, $10$ or $100$, the wallclock time per iteration does not suffer much, certainly not linearly in $K$. What fundamentally limits $K$ for us is GPU memory. We will make sure to discuss these issues more thoroughly in the paper.
>
> To provide more clarity on the role of $K$, we have systematically varied $K$ in the delayed-addition task that was suggested to us by Rev. 3Uai (Figure 2 of the one-page PDF). As expected, SOFO gets better with increasing $K$.
>
> You also raised concerns about SOFO's ability to train much larger networks (e.g. size 1000 as for FORCE). We trained networks of size up to $S=8000$ in the context of the 3-bit flip-flop task, but could have gone higher given that $S=8000$ only used half of our GPU memory. However, $S=8000$ worked well here because, just like FORCE, we only optimized the readout weights on this task (not the full recurrent weight matrix). Optimizing the full recurrent matrix would have the same runtime complexity, but a larger memory overhead given that we would need to store $K$ (~100) parameter tangents of size $S^2$. This is not an unsurmountable obstacle, as these tangents are random and can in principle be evaluated lazily (e.g. recomputed on demand based on a random seed). Sadly this would require more software engineering than we can pull off in this short rebuttal period.
>
> Finally, SOFO is indeed readily applicable to settings where a loss function is not defined for all time steps (would FORCE work in such a setting?) -- this is discussed on L.91, and the one-shot classification example of Figure 3 is an example of this.
>
> We will address all your minor comments in the revision.

---

> > ### Comment · Reviewer_Z9Qd · 2024-08-07
> >
> > Wow! Amazing rebuttal, amazing responses, and amazing experiments. Both of my concerns are well addressed, thank you very much for it. I strongly support the acceptance of this work, which I am looking forward to using myself!

---

> > > ### Comment · Reviewer_Z9Qd · 2024-08-07
> > >
> > > Also, please add the details on the batch FORCE approach to an Appendix. I believe this is another important and novel contribution.

---

> ### Author Response · Authors · 2024-08-08
>
> Thank you very much for constructively engaging with our work!
>
> We will of course add the details of batched FORCE in an Appendix.

---

### Official Review · Reviewer_mSGm · 2024-07-12

**Soundness:** 3
**Presentation:** 3
**Contribution:** 2
**Rating:** 5
**Confidence:** 3

**Summary:**

In this work, the authors applied a second-order optimization with an approximated Hessian to train vanilla recurrent neural networks (RNNs). The authors have shown that the weight update using a random subspace approximation of Hessian can be implemented onto a GPU in a memory-efficient manner by using the Jacobian-vector product. They applied the algorithm to Lorenz attractor, adaptive Kalman filter, one-shot learning, and motor reaching tasks, and demonstrated that the proposed algorithm achieves better performance than the Adam optimizer while achieving comparable wall-clock time.

**Strengths:**

The manuscript is written clearly and technically sound. The authors implemented the proposed algorithm in several tasks including biologically-motivated ones. The presented results clearly demonstrate the advantage of the proposed algorithm over the Adam optimizer in their experimental setting.

**Weaknesses:**

The contribution to both machine learning and neuroscience are rather weak. The Gauss-Newton method with random subspace estimation was studied previously (Gower et al., NeurIPS 2019; Cartis et al., arXiv, 2022), and efficient GPU implementation of forward gradient-based optimization methods was also developed before (e.g., Baydin et al., arXiv, 2022). While its application to RNN is novel to the best of my knowledge, the authors did not provide any analytical results, for instance, on how sketching of Hessian estimation and noise in RNN dynamics corrupt the weight updates. The implementation did not take advantage of the simplicity of RNN architecture either.

Regarding contribution to neuroscience, while the authors focus on vanilla RNNs due to the biological implausibility of gated recurrent units and LSTMs, their proposed learning algorithm also lacks biological plausibility. Although they mention a “striking resemblance with those recorded in the motor cortex,” this resemblance is not quantitatively evaluated. Additionally, it remains unclear if the proposed algorithm converges to a solution more similar to those implemented in the motor cortex compared to previous optimization algorithms.

While the authors motivate this work with ‘reverse engineering’ approach for studying neural circuits, its potential contribution to neuroscience is not explained clearly. It would be helpful if the authors could explain what kind of biological questions they aim to address with this algorithm.

**Questions:**

Evaluations in the manuscript were mostly conducted in networks with a small number of neurons (N=64) and a small time-bin (2ms). Do you expect the proposed approach to also benefit larger networks with coarser discretization of time?

**Limitations:**

Related to the question above, I believe some comments on the hyperparameter choices would strengthen the clarity of the paper.

---

> ### Author Rebuttal · Authors · 2024-08-07
>
> Thank you for your thoughtful review; we are glad you found the paper clearly written and technically sound and that you found our experiments convincing. In the main rebuttal above, we have addressed the concerns common to all reviews, including your comment on the effect of network size. Here we would like to address your other concerns.
>
> Regarding your novelty concerns: to the best of our knowledge, we are the first to really take advantage of GPU parallelism to implement random-subspace Gauss-Newton in an efficient manner (where previous works on second-order sketching had used (i) sequentially evaluated finite-differences approximations of directional derivatives (Cartis et al., 2022), or (ii) memory-expensive reverse-mode differentiation (Gower et al.)). We find that parallelising Jacobian-vector products is really key to outperforming Adam (or other first-order optimizers) on _wall-clock time_ in all our experiments. We achieve this through a custom implementation of _batched_ forward-mode differentiation, which was in itself a challenge as it is not natively available in common GPU frameworks (e.g. pytorch). We will of course release this library on github together with the paper.
>
> Our work also improves substantially on Baydin et al.: their work on Forward-Gradient Descent (FGD) only used a single tangent vector to estimate the gradient, and therefore only required standard (non-batched) forward-mode AD. What we do instead is compute $K$ directional derivatives in parallel, leading to reduced stochasticity in our gradient and GGN estimates. As our experiments show, even in the easiest task of Figure 1B (Lorenz attractor), FGD fails; in fact, it even fails when extended to the large $K$ setting, and SOFO rescues this by incorporating the GGN matrix. In summary, while previous literature had indeed laid the groundwork for our study, SOFO uniquely expands and combines these previous strands to yield an efficient new method for training RNNs.
>
> Regarding analytical results: we agree that this was not the focus of our work; for this, we rely on SOFO inheriting the theoretical guarantees previously derived for sketching algorithms in general, not just for RNNs (e.g. Cartis et al., ICML 2020), and focus on showing empirical performance.
>
> Regarding neural activity in the network trained on the motor task in Section 4.4 bearing a “striking resemblance with those recorded in the motor cortex”, we agree that this statement relies on mere visual inspection/comparison of firing rate time series (e.g. see Churchland et al., 2012 for examples of monkey M1 PSTHs) and was not made quantitative. We believe this is perhaps better left to a future study, as doing such comparisons carefully takes significant time and effort, and is not the focus of our paper. The focus here is on establishing the methodology that will enable such studies in future. We are in fact excited about this and have begun to analyse SOFO-trained RNN models of motor cortex; given our preliminary analyses, all we feel comfortable saying at this stage is that our trained RNNs capture at least two salient features of motor cortex dynamics prior to and during reaching: (i) movement-related activity exhibits consistent state-space rotations revealed by jPCA (Churchland et al., 2012) and (ii) is largely orthogonal to preparatory activity (Elsayed et al., 2016).
>
> Regarding the type of biological inquiries that are enabled by the training and subsequent dissection of RNNs, we would like to refer you to the great review by Omri Barak on “Recurrent neural networks as versatile tools for neuroscience research” (Cur. Op. Neurobiol., 2017). This type of approaches have already been influential (e.g. Mante, Sussillo et al, Nature 2013; Sussillo et al., Nat. Neurosci. 2015) and have helped understand of how brains “compute through dynamics”. Since these pioneering contributions, the use of trained RNNs has become increasingly common in neuroscience, see e.g. Driscoll et al., Nat Neurosci 2024 for the more recent developments. We will add these references to our paper. We believe that by significantly accelerating the training of RNNs, SOFO can enable faster progress in this space. In particular, there are a number of computations which brains perform, such as (active) learning or contextual adaptation, that rely on integrating inputs over long behavioural time scales. To the best of our knowledge, few attempts have been made to train RNNs to chart the space of network solutions to these long-horizon problems, and progress here will rely on memory-efficient training algorithms such as SOFO which are capable of handling long horizons.
>
> > their proposed learning algorithm also lacks biological plausibility
>
> Just to clarify: our algorithm is not meant as a model of biological learning; it is meant as an optimization tool to find plausible network solutions to various problems which brains solve.
>
> > Do you expect the proposed approach to also benefit larger networks with coarser discretization of time?
>
> For larger networks, see our overall rebuttal. Regarding time discretisation: our one-shot classification task of Figure 3 is an example where we did not have an explicit "dt" term in the dynamics (equivalently, the dynamics did not incorporate an explicit leak term), i.e. it is an example of coarse discretization. More generally though, we believe that how coarsely one can afford to discretize time does not depend on the optimizer being used, but on the task itself: if, say, a task has trials that last for a second of biological time, and where the millisecond-scale details of the network output influence the task loss, then at least 1000 time bins must be used; in other words, we would expect all optimizers to be equally affected by a poor choice of time discretization that is not suitable for the task at hand.

---

> > ### Comment · Reviewer_mSGm · 2024-08-12
> >
> > Thank you so much for the revision.
> >
> > I still have reservations about its potential contribution to neuroscience. I find it concerning to use an RNN trained with a learning rule that completely lacks biological plausibility to generate hypotheses about how the brain works, particularly given the vast potential solution space. Additionally, if training RNNs is so difficult, it raises the question of why the brain would rely on such a circuitry. Could it be that we are overlooking something more fundamental?
> >
> > That said, I believe this is a solid piece of work, and I have no technical objections to its acceptance.

---

### Official Review · Reviewer_535W · 2024-07-16

**Soundness:** 4
**Presentation:** 3
**Contribution:** 3
**Rating:** 8
**Confidence:** 4

**Summary:**

In this work, the authors develop a method to train recurrent neural networks with biological plausibility constraints, which are required for neuroscience applications, and where standard BPTT doesn't tend to work well. They use forward-mode gradient calculation (RTRL) that makes the memory requirement independent of length of input sequence. To mitigate the high compute cost of RTRL, they combine it with a second order optimizer that is parallelizable on a GPU and makes the overall training wall-clock time comparable to BPTT. The second order optimizer uses a randomized method to calculate the Generalized Gauss-Newton matrix. They demonstrate their method in a number of neuroscience inspired tasks.

**Strengths:**

The overall problem domain is well defined and well motivated. Neuroscience does have constraints in terms of network architectures that can be used, and so it makes sense to improve plain backprop even if it comes at an additional computational cost (which in this particular approach is not an issue).

Their use of forward mode to handle commonly occurring long input sequences makes sense. The choice of GGN updates with a randomized algorithm is also reasonable.

Overall their paper is well written and structured. The the experiments are well chosen.

**Weaknesses:**

I would have liked to see a more detailed ablation study to understand the contribution of the various components of the algorithm (e.g. choice of K, size of network)
And perhaps a LSTM/GRU baseline just to understand the gap between vanilla RNNs and architectures used in ML (although this is not critically relevant for this study).

I would also have liked a bit more discussion into why vanilla RNNs are the right architecture for neuroscience, since at best they would be crude simplifications of the complex dynamics of biological neurons. And I would expect its rate-based representations don't provide a good model for brain areas that tend to use more spike time based representations.

**Questions:**

- It's not clear why a stochastic RNN was chosen for the motor task.
- In line 90, the authors say "In RNN applications, i indexes both time bins and RNN output dimensions" which doesn't sound quite right.

**Limitations:**

The authors discuss the limitations clearly.

---

> ### Author Rebuttal · Authors · 2024-08-07
>
> We thank you for your thoughtful review; we are glad you found the paper well written and structured. In the main rebuttal above, we have addressed the concerns common to all reviews, including a more systematic study of the effect of $K$ (number of random parameter tangents used to sketch the GGN update) on SOFO's performance relative to Adam. Given the time constraints of this one-week rebuttal period, we have performed this study in the context of a new task (delayed addition) that was suggested to us by Rev. Z9Qd, but we will progressively be adding similar experiments for some of the other tasks we used in the original submission.
>
> We haven't had the time to do comparisons with LSTMs/GRUs, but we can definitely get started on this after the rebuttal deadline if you think this would add value. We expect that for LSTMs/GRUs, SOFO will have a reduced advantage over Adam, and that those networks will perform overall better than their vanilla counterparts.
>
> Regarding the choice of vanilla RNNs as models for neuroscience: as you rightly point out, neuroscience isn't yet at a point where there is a consensus on _the right architecture_ to describe the dynamics of neuronal networks. In this context, many studies have used vanilla RNNs as an Occam-razor-like compromise: simple integration of input firing rates appears to be what neurons do to a decent first order approximation, and until we have more clarity on whether and how neurons might be implementing more sophisticated mechanisms such as forget/input/output gates, sticking to simpler models appears sensible to many in the field. Thus, we see SOFO as relevant to neuroscience primarily because of the pervasiveness of vanilla RNNs in the field, not necessarily because they are the most accurate models. Indeed, we agree that these simpler models are crude, and it will be important to develop methods for training more biologically realistic (e.g. spiking) variants as they become more mainstream. We will add a paragraph of discussion along these lines, thanks for prompting us to do so.
>
> > not clear why a stochastic RNN was chosen for the motor task
>
> We wanted to have at least one example of a stochastic RNN to cover more diversity of dynamics, as stochastic vanilla RNNs are also common in the literature, for example to study across-trial variability and its relationship with computation (e.g. Echeveste et al., 2020).
>
> > In line 90, the authors say "In RNN applications, i indexes both time bins and RNN output dimensions" which doesn't sound quite right.
>
> What we meant was that SOFO optimizes general loss functions that are summed over both RNN output dimensions (e.g. target behaviour of the RNN in a certain number $D$ of output channels) and over time. We could have used two distinct indices, say $j$ and $t$, and have a triple sum in Eq. (1), but since these two dimensions are completely exchangeable from the perspective of the SOFO algorithm, we thought we might as well collapse them into a single index ($i$, which then runs from 1 to DT). Another reason we simplified the notation this way is because SOFO should also be applicable to feedforward (non-recurrent) networks for which there is simply no time index -- the way Eq 1 is written, it still applies to such cases. We will clarify this in the text.

---

> > ### Comment · Reviewer_535W · 2024-08-12
> >
> > I thank the authors for their responses. The new results look very promising.
> >
> > Reg. LSTM/GRU comparison, I was very curious about the potential for using this method in ML models, and perhaps a performance upper bound. I think this could make the study relevant to the ML community as well and strengthen the paper, but I’ll leave the decision to do this up to the authors.
> >
> > Thank you for the discussion about vanilla RNNs, agree and would be useful to have this briefly in the paper.
> >
> > I’m satisfied with all the other responses. Overall I think this is a very good paper and I find it very exciting. So I’ll increase my score further.

---

### Author Rebuttal · Authors · 2024-08-07

We thank all reviewers for the time and effort they put into providing thoughtful reviews. We have addressed most of their concerns in reviewer-specific rebuttals, and here is an executive summary:

- We have added a comparison to FORCE learning, on a new task (3-bit flip flop) -- see Figure 1 of the one-page PDF; SOFO performs better than FORCE in this task.  We encourage all reviewers to take a quick look at our rebuttal to Rev. Z9Qd (who suggested this addition), where we discuss in what ways SOFO is not just quantitatively better than FORCE in this case, but also very complementary: it allows training networks initialized in a non-chaotic state (where FORCE relies on chaos), and is capable of training much smaller networks on this challenging task, using more realistic nonlinearities such as ReLU (where FORCE is largely restricted to tanh or other symmetric activation functions due to its inability to optimize single-neuron biases).
- In the context of this 3-bit flip-flop task, we have trained networks of size up to 8000, i.e. two orders of magnitude larger than the networks we trained in the paper.
- We have added results for a delayed addition task suggested by Rev. Z9Qd (Figure 2 of the one-page PDF). There, SOFO again outperforms Adam. In this task, we have also systematically varied the number of tangents $K$ (size of the GGN / gradient sketch) to provide more clarity as to the role of this hyper-parameter. With larger $K$, both SOFO's estimates of the gradient and GGN matrix get more accurate, leading to better performance (see also Appendix A.6). Empirically, we show that for RNN applications, SOFO works well when the ratio of $K$ to the number of parameters $P$ is as low as $1$%, which is one of the reasons why SOFO is so pratical in this setting. When using larger networks (as we did in the flip-flop task), GPU memory forces us to use a smaller $K/P$ ratio, and SOFO does suffer from this: training the network of size $S=8000$ requires more iterations than for $S=1000$; we will more clearly lay out this limitation of SOFO in our revision.
- We have also added a comparison between Adam and SOFO on an MLP-Mixer architecture (~$10^5$ parameters) trained on CIFAR10 (as suggested by Rev. 3Uai) -- see Figure 3 of the one-page PDF. This is a fairly easy target for Adam due to the presence of skip connections and layer-norm, and with the limits on $K$ imposed by GPU memory, the best SOFO does here is to nearly match Adam's performance. Note that this shallow network setup is not where we expect SOFO to be the most useful; its primary strength is in fixing the $\mathcal{O}(T)$ memory complexity of backpropagation through time in RNNs, making it $\mathcal{O}(T^0)$, enabling training in settings where backprop with either large networks or large data batches is not feasible for a given long time horizon.
- Prompted by Rev. 3Uai, we have studied the relationship between SOFO and Real-Time Recurrent Learning (RTRL). RTRL turns out to be mathematically equivalent to a particular limit of SOFO: that of (i) not using curvature information and (ii) using a full, identity basis of parameter tangents (as opposed to a small number of random tangents). This is a lot more memory intensive, as we demonstrate by running memory profiling for RTRL; we find that we can only afford to run RTRL provided that we also dramatically reduce the data minibatch size.

We hope that the reviewers will find, as we do, that these additions strengthen our paper.

---

### Decision · Program_Chairs · 2024-09-25

**Decision:**

Accept (spotlight)

**Comment:**

This paper suggests using second-order optimization, essentially a type of quasi-Newton method, for “vanilla” RNNs as used in neuroscience. While the method itself is not that new, the authors introduce an efficient GPU-based parallelization that makes runtimes comparable to those achieved with standard gradient-based techniques, but with largely superior performance. This is indeed an important methodological contribution which shifts the focus from improving architectures to improving training algorithms, following a recent trend in ML. While initially there were some concerns about novelty (see above), biological plausibility, or size of the models used (well addressed in the rebuttal), overall this manuscript received a fairly enthusiastic response from most referees, with all four referees ultimately on the accept side, which I support.